# RhoGDI phosphorylation by PKC promotes its interaction with death receptor p75^NTR to gate axon growth and neuron survival

Ajeena Ramanujan [1,7], Zhen Li [2,7], Yanchen Ma [3,4], Zhi Lin[2] & Carlos F Ibáñez [1,3,4,5,6 ✉]

## Abstract

**How receptors juggle their interactions with multiple downstream effectors remains poorly understood. Here we show that the outcome of death receptor p75^NTR signaling is determined through competition of effectors for interaction with its intracellular domain, in turn dictated by the nature of the ligand. While NGF induces release of RhoGDI through recruitment of RIP2, thus decreasing RhoA activity in favor of NFkB signaling, MAG induces PKC-mediated phosphorylation of the RhoGDI N-terminus, promoting its interaction with the juxtamembrane domain of p75^NTR, disengaging RIP2, and enhancing RhoA activity in detriment of NF-kB. This results in stunted neurite outgrowth and apoptosis in cerebellar granule neurons. If presented simultaneously, MAG prevails over NGF. The NMR solution structure of the complex between the RhoGDI N-terminus and p75^NTR juxtamembrane domain reveals previously unknown structures of these proteins and clarifies the mechanism of p75^NTR activation. These results show how ligand-directed competition between RIP2 and RhoGDI for p75^NTR engagement determine axon growth and neuron survival. Similar principles are likely at work in other receptors engaging multiple effectors and signaling pathways.**

**Keywords** RhoA; Myelin Associated Glycoprotein; Nerve Growth Factor; RIP2
**Subject Categories** Membranes & Trafficking; Neuroscience; Signal Transduction

## Introduction

Single-pass transmembrane receptors typically engage multiple signaling pathways simultaneously. While catalytic receptors with kinase activity activate downstream targets through phosphorylation, non-catalytic receptors elicit intracellular signaling events through the recruitment and release of downstream effector molecules to and from the receptor intracellular domain. Non-catalytic receptors have the capacity to interact with a variety of downstream effectors, each leading to different signaling pathways that often control diverse outcomes in the cell. An overabundance of pathway-centered studies have left a knowledge gap with respect to the mechanisms by which these receptors engage different signaling pathways concurrently, and the conditions that allow some pathways to overtake others.

The p75 neurotrophin receptor (p75^NTR) is a non-catalytic transmembrane receptor with a capacity to engage several different signaling pathways, including the RhoA pathway, that regulates cytoskeleton dynamics and cell motility (Yamashita et al, 1999, 2002; Yamashita and Tohyama, 2003), the NF-kB pathway, which regulates gene expression and promotes cell survival (Carter et al, 1996; Kisiswa et al, 2018), and the JNK pathway, which can lead to activation of caspase-3 and apoptosis (Casaccia-Bonnefil et al, 1996; Friedman, 2000). Like other members of the TNF receptor superfamily, p75^NTR contains a 6-helix globular domain in its intracellular region known as the "death domain", herein abbreviated DD (Liepinsh et al, 1997). The DD is linked to the transmembrane domain by a 60 residue-long, unstructured, juxtamembrane region, herein termed JXT (Liepinsh et al, 1997). The extracellular domain of p75^NTR can interact with a variety of ligands, including the neurotrophins, such as nerve growth factor (NGF), the beta-amyloid peptide, and various viral capsid proteins (Roux and Barker, 2002; Underwood and Coulson, 2008). In addition, several myelin-derived components, such as myelin-associated glycoprotein (MAG), require p75^NTR as a signaling component in a receptor complex containing Nogo receptor (NgR) and Lingo-1 as ligand binding co-receptors (Wang et al, 2002a; Wong et al, 2002; Mi et al, 2004). It has been suggested that MAG does not interact directly with p75^NTR (Yamashita et al, 2002), but is still unclear whether it makes direct contacts with p75^NTR in the multimeric receptor complex. Previous studies have shown that different p75^NTR ligands can elicit distinct signaling outcomes, but it remains unclear how the receptor integrates the signals induced by concomitant exposure to more than one ligand, a situation that is more likely to resemble physiological conditions. NGF binding to

[1]Department of Physiology and Life Sciences Institute, National University of Singapore, 117456 Singapore, Singapore. [2]Tianjin Key Laboratory of Function and Application of Biological Macromolecular Structures, School of Life Sciences, Tianjin University, Tianjin 300072, China. [3]Peking University School of Life Sciences, PKU-IDG/McGovern Institute for Brain Research, Peking-Tsinghua Center for Life Sciences, 100871 Beijing, China. [4]Chinese Institute for Brain Research, Life Science Park, 102206 Beijing, China. [5]Department of Neuroscience, Karolinska Institute, Stockholm 17177, Sweden. [6]Stellenbosch Institute for Advanced Study, Wallenberg Research Centre at Stellenbosch University, Stellenbosch 7600, South Africa. [7]These authors contributed equally as first authors: Ajeena Ramanujan, Zhen Li. ✉E-mail: carlos.ibanez@pku.edu.cn

p75$^{NTR}$ recruits the RIP2 adaptor to its DD, linking to the NF-kB pathway (Khursigara et al, 2001). Recruitment of RIP2 displaces Rho GDP-dissociation inhibitor (RhoGDI) from the receptor DD, leading to reduced RhoA activity (Lin et al, 2015). On the other hand, MAG has been shown to enhance binding of RhoGDI to p75$^{NTR}$ and increase RhoA activity (Yamashita et al, 2002; Yamashita and Tohyama, 2003), which in neuronal cells leads to growth cone collapse and stunted axon growth (Wang et al, 2002a; Wong et al, 2002; Park et al, 2010). In cerebellar granule neurons, MAG can also induce cell death by apoptosis (Fernández-Suárez et al, 2019). The mechanism by which MAG enhances the interaction of RhoGDI with p75$^{NTR}$ is unclear. Neither is it known whether such process has any effect on the interaction of RIP2 with the receptor and NF-kB activity or cell survival. Resolving these questions is important for our understanding of receptor function, as neurons expressing p75$^{NTR}$ are likely to encounter both NGF and MAG in vivo.

Although initially described as a negative regulator of Rho GTPases, RhoGDI does also exert positive functions in the GTPase cycle, including protein stabilization of Rho family GTPases as well as their recruitment to plasma membrane receptors, such as p75$^{NTR}$ and TROY (Dovas and Couchman, 2005; Hodge and Ridley, 2016; Lin et al, 2015; Lu et al, 2013). Engagement of the RhoGDI/RhoA complex with p75$^{NTR}$ results in the release of RhoA, its insertion in the plasma membrane and subsequent activation (Yamashita and Tohyama, 2003; Lin et al, 2015). RhoGDI has been shown to interact with the DD of p75$^{NTR}$ through its C-terminal Ig-like domain (Lin et al, 2015; Yamashita and Tohyama, 2003). Our group has previously solved the NMR structure of this complex showing that the C-terminal domain (CTD) of RhoGDI binds to the p75$^{NTR}$ DD without affecting DD-DD homodimerization (Lin et al, 2015). Although RhoGDI interacts with RhoA and the p75$^{NTR}$ DD through non-overlapping interfaces, biochemical and structural studies have indicated that binding of the RhoGDI CTD to the p75$^{NTR}$ DD reduces RhoGDI affinity for RhoA, possibly by propagation of conformational changes to the RhoGDI/RhoA binding interface (Lin et al, 2015). Previous studies have indicated that Protein Kinase C (PKC) can phosphorylate RhoGDI on its N-terminal domain (NTD) leading to release and activation of RhoA (Dovas et al, 2010; Knezevic et al, 2007; Sabbatini and Williams, 2013), but whether such mechanism is implicated in p75$^{NTR}$ signaling has not been investigated. PKC has been shown to mediate some of the inhibitory effects of myelin-derived components on axonal regeneration (Sivasankaran et al, 2004), but whether this is involved in MAG signaling through p75$^{NTR}$ to RhoGDI and RhoA is unknown.

In this study, we set out to investigate the role of PKC in RhoGDI recruitment to p75$^{NTR}$ as well as the ability of the receptor to regulate RhoA activity in response to MAG and NGF. Through a variety of studies in cell lines and cultured primary neurons, we found that stimulation of cells with MAG induced PKC-mediated phosphorylation of Ser$^{34}$ in the NTD of RhoGDI, previously thought not to be involved in its interaction with p75$^{NTR}$ (Yamashita and Tohyama, 2003). This strengthened RhoGDI binding to the receptor and enhanced RhoA activity, while at the same time prevented RIP2 binding, diminished activation of NF-kB and profoundly affected axonal growth and neuron survival. These findings were validated through both pharmacological and genetic studies using a series of RhoGDI and p75$^{NTR}$ mutants. A novel NMR solution structure of the NTD of RhoGDI in complex with p75$^{NTR}$ JXT domain is also presented.

## Results

### PKC-mediated phosphorylation of RhoGDI at Ser$^{34}$ enhances RhoGDI binding to p75$^{NTR}$ and activation of RhoA

In cerebellar granule neurons (CGNs), baseline levels of RhoGDI binding to p75$^{NTR}$ were eliminated upon treatment with the specific PKC inhibitor Gö6976, while PKC activation with phorbol myristate acetate (PMA) greatly enhanced RhoGDI/p75$^{NTR}$ interaction (Fig. 1A), suggesting that PKC-mediated RhoGDI phosphorylation promotes its binding to the intracellular domain of the receptor. In line with this, treatment of CGNs with MAG promoted RhoGDI binding to p75$^{NTR}$ in a PKC-dependent manner, as it could be abolished by treatment with Gö6976 (Fig. 1B,C). Previous studies indicated that RhoGDI phosphorylation by PKC results in the release and activation of RhoA, but differed in the proposed target residues of phosphorylation, with both Ser$^{34}$ (Dovas et al, 2010) and Ser$^{96}$ (Knezevic et al, 2007; Sabbatini and Williams, 2013) put forward as alternatives. We generated phosphorylation-mimetic mutants of RhoGDI with aspartic acid replacing either serine residue (S34D and S96A, respectively) and assessed their interaction with p75$^{NTR}$ in NIH3T3 cells (which do not express endogenous p75$^{NTR}$) transfected with expression plasmids for p75$^{NTR}$ and Flag-tagged RhoGDI constructs. We found that S34D significantly enhanced the binding of RhoGDI to the receptor, while mutation to alanine (S34A) greatly reduced RhoGDI binding to p75$^{NTR}$ (Fig. 1D,E). On the other hand, replacement of Ser$^{96}$ to Asp (S96D) or Ala (S96A) had no effect on the interaction of RhoGDI with p75$^{NTR}$ (Fig. 1F). In contrast to wild type (WT) RhoGDI, neither S34A nor S34D mutants were affected in their interaction with p75$^{NTR}$ by either inhibition (Gö6976) or activation (PMA) of PKC (Fig. 1G,H), suggesting that Ser$^{34}$ is the main, if not the only, residue whose PKC-mediated phosphorylation promotes RhoGDI binding to the receptor. We also found that the cleavage-resistant V246N mutant of p75$^{NTR}$ (Coulson et al, 2008; Underwood et al, 2008) interacted with RhoGDI at comparable levels to the wild-type receptor, both under baseline conditions and upon PMA stimulation (Fig. 1I), indicating that p75$^{NTR}$ cleavage is not necessary for PKC-stimulated recruitment of RhoGDI.

Next, we examined the effects of PKC-mediated phosphorylation of RhoGDI on the regulation of RhoA activation, as assessed by the level of RhoA-GTP. In this assay, absorbance at 490 nm measures the active GTP-bound RhoA in the cell lysates that have bound to the Rhotekin RhoA binding domain immobilized to 96-well plates. Introduction of RhoGDI S34D phospho-mimetic mutant by lentiviral transduction into wild-type CGNs significantly enhanced RhoA activation in wild-type CGNs but not in neurons lacking p75$^{NTR}$ (Fig. 1J). In fact, overexpression of either wild type or S34D mutant RhoGDI reduced RhoA-GTP levels in neurons lacking p75$^{NTR}$ (Fig. 1J), supporting the notion that RhoGDI requires p75$^{NTR}$ to function as enhancer of RhoA activation, behaving as inhibitor in its absence. Interestingly, the effect of PMA on RhoA activation also required p75$^{NTR}$ as it was abolished in CGNs lacking p75$^{NTR}$ (Fig. 1K), suggesting that stimulation of PKC activity is not sufficient and binding of RhoGDI to p75$^{NTR}$ is also required for activation of RhoA. Treatment of CGNs with PMA increased RhoA-GTP levels in neurons expressing wild-type RhoGDI, in agreement with previous findings (Dovas et al, 2010),

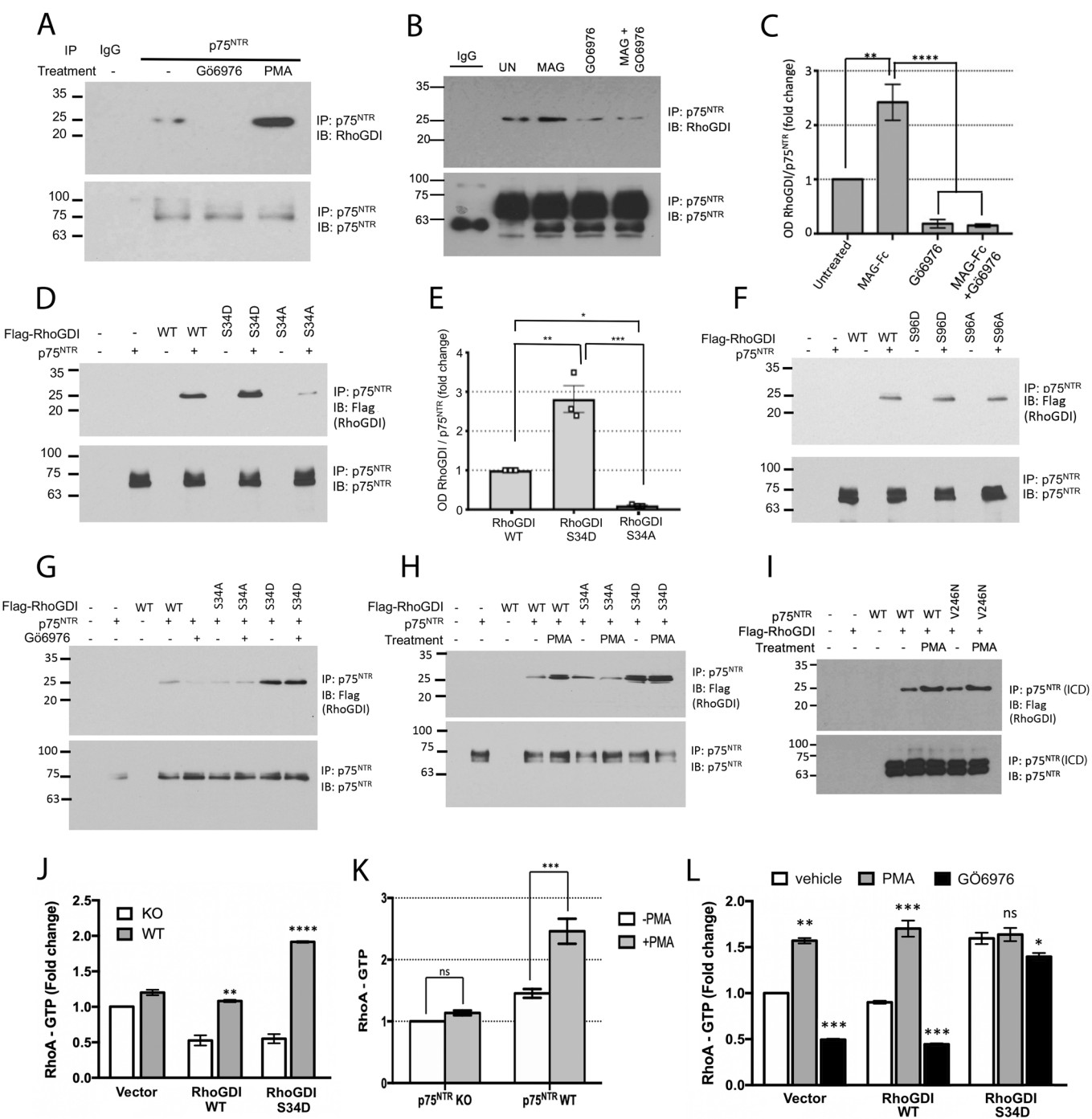

but did not elevate RhoA-GTP levels further in neurons that received the S34D RhoGDI mutant (Fig. 1L). Conversely, inhibition of PKC by treatment with Gö6976 reduced RhoA-GTP levels in neurons that expressed wild-type RhoGDI, but RhoA-GTP levels were unaffected in neurons overexpressing the S34D RhoGDI mutant (Fig. 1L). Together these results indicate that PKC-mediated phosphorylation of RhoGDI at Ser[34] enhances RhoGDI interaction with p75[NTR] and promotes activation of RhoA, while Ser[96] phosphorylation is dispensable for RhoGDI binding to the receptor.

## Enhanced binding of RhoGDI to p75[NTR] through phosphorylation of Ser[34] induces phosphorylation of Ser[96] leading to RhoA dissociation and activation

We investigated the effects of phosphorylation-mimetic RhoGDI mutants on the interaction between RhoGDI and RhoA as well as RhoA GTPase activity. In NIH3T3 cells, which lack endogenous p75[NTR] expression, neither RhoGDI[S34D] nor RhoGDI[S34A] had any effect on the interaction between RhoGDI and RhoA (Fig. 2A). In contrast, S96D greatly reduced this interaction, while S96A

**Figure 1.  PKC-mediated phosphorylation of RhoGDI at Ser[34] enhances RhoGDI binding to p75[NTR] and activation of RhoA.**

(A) Representative immunoblots of p75[NTR] immunoprecipitation of protein from cultured P7 p75[NTR] wild-type CGNs treated with either GO6976 (1 µM) or PMA (10 ng/ml) overnight and then probed with antibodies to RhoGDI and p75[NTR]. Note the diminished interaction of RhoGDI with p75[NTR] in the presence of GO6976. Immunoblotting of lysates (Fig. EV2) with phospho-serine PKCα substrate antibody confirms role of PMA in activating PKC. IP, immunoprecipitation; IB, immunoblotting. Western blots of whole cell lysates are shown in Appendix Fig. S1A. (B) Representative immunoblots of p75[NTR] immunoprecipitation of protein from cultured P7 p75[NTR] wild-type CGNs treated with either MAG (25 µM) or GO6976 (1 µM) or a combination of both MAG and GO6976 for 30 min, and then probed with antibodies to RhoGDI and p75[NTR]. Note the diminished interaction of RhoGDI with p75[NTR] in the presence of Gö6976. UN, untreated. Western blots of whole cell lysates are shown in Appendix Fig. S1B. (C) Quantification of RhoGDI in p75[NTR] immunoprecipitates from P7 p75NTR wild-type CGNs treated with either MAG (25 µM) or Gö6976 (1 µM) or a combination of both MAG and GO6976 shown in (B). Densitometric values are plotted as mean + SEM from three independent experiments and normalized to the untreated control (**$p < 0.01$, ****$p < 0.001$ by one-way ANOVA followed by Tukey's multiple comparisons test). (D) Representative immunoblots of p75[NTR] immunoprecipitation of protein from NIH3T3 cells cotransfected with p75[NTR] along with either RhoGDI wild type, S34D or S34A mutant. Western blots of whole cell lysates are shown in Appendix Fig. S1C. (E) Quantification of RhoGDI in p75[NTR] immunoprecipitates from NIH-3T3 cells after transfection with p75[NTR] along with the indicated RhoGDI constructs. Densitometric values are plotted as mean + SEM from three independent experiments and normalized to the WT control (*$p < 0.01$, **$p < 0.001$, ***$p < 0.001$ by one-way ANOVA followed by Tukey's multiple comparisons test). (F) Representative immunoblots of p75[NTR] immunoprecipitation of protein from NIH3T3 cells cotransfected with p75NTR along with either RhoGDI wild type, or Ser96 single mutants of RhoGDI. Western blots of whole cell lysates are shown in Appendix Fig. S1D. (G,H) Representative immunoblots of p75[NTR] immunoprecipitation of protein from NIH3T3 cells cotransfected with p75NTR along with either RhoGDI wild type, S34D or S34A mutant. Twelve hours post transfection, the cells were treated with GO6976 (G) or PMA (H). Western blots of whole cell lysates are shown in Appendix Fig. S1E, F. (I) Representative immunoblots of p75[NTR] immunoprecipitates from NIH3T3 cells transfected with p75[NTR] wild type or a γ-secretase mutant variant p75[NTR] V246N and then probed with antibodies flag for RhoGDI and for p75[NTR]. p75[NTR] was pulled down using ANT-011 antibody raised against the intracellular region (ICD) of p75[NTR]. Western blots of whole cell lysates are shown in Appendix Fig. S1G. (J) Quantification of active RhoA in cultured P7 KO or WT CGNs transduced with lentivirus expressing either RhoGDI wild type or RhoGDI S34D mutant. Mean ± SEM of reading at 490 nm from three independent experiments performed in triplicate wells is shown (**$p < 0.01$, ****$p < 0.0001$ by two-way ANOVA followed by Tukey's multiple comparison test). (K) Quantification of active RhoA in cultured P7 KO or WT CGNs with or without treatment with PMA (10 ng/ml). Mean ± SEM of reading at 490 nm from 3 independent experiments performed in duplicate wells is shown (***$p < 0.001$ two-way ANOVA followed by Tukey's multiple comparison test). (L) Quantification of active RhoA after treatment with or without Gö6976 (1 µM) or PMA (10 ng/ml) in cultured P7 WT CGNs transduced with lentivirus expressing either RhoGDI wild type or RhoGDI S34D mutant. Mean ± SEM of reading at 490 nm from 3 independent experiments performed in duplicate wells is shown (*$p < 0.05$, **$p < 0.01$, ***$p < 0.001$, two-way ANOVA followed by Tukey's multiple comparison test). Source data are available online for this figure.

enhanced it (Fig. 2B,C), suggesting that phosphorylation of Ser[96] leads to dissociation of RhoA from RhoGDI. Baseline RhoA/RhoGDI binding was greatly reduced by introduction of p75[NTR] in NIH3T3 cells and was completely abolished if in addition cells also expressed the S34D RhoGDI mutant (Fig. 2D last lane). Interestingly, interaction of RhoA with RhoGDI[S34D] could be rescued by concomitant mutation of Ser[96] to alanine (S34D/S96A double mutant, Fig. 2D), supporting the idea that, by enhancing RhoGDI binding to p75[NTR], phosphorylation at Ser[34] triggers phosphorylation at Ser[96] leading to dissociation of RhoA from RhoGDI. In order to test this further, we generated phospho-specific antibodies against phospho-Ser[96] (P-Ser[96]). In wild-type CGNs, overexpression of RhoGDI S34D mutant, but not S34A, induced the phosphorylation of Ser[96] (Fig. 2E). In these neurons, basal phosphorylation of Ser[96] could be enhanced by treatment with MAG, while inhibition of PKC with Gö6976 abolished all Ser[96] phosphorylation (Fig. 2F). In CGNs isolated from p75[NTR] knock-out (KO) mice, baseline phosphorylation at RhoGDI Ser[96] could only be detected upon reintroduction of p75[NTR] through lentiviral transduction, and PMA strongly induced this phosphorylation in the presence, but not in the absence, of the receptor (Fig. 2G). Next, we introduced plasmids to express Flag-tagged wild type (WT), S34A, or S34D mutant RhoGDI into wild-type CGNs and assessed Ser[96] phosphorylation in anti-Flag immunoprecipitates after treatment with MAG or vehicle (Fig. 2H). MAG induced phosphorylation of Ser[96] in neurons expressing WT RhoGDI but not in the S34A mutant, while Ser[96] phosphorylation in the S34D mutant was unaltered by MAG treatment (Fig. 2H). Finally, transfection of S96A RhoGDI mutant reduced, but S96D enhanced, RhoA activation in both wild type and p75[NTR] KO CGNs (Fig. 2I), in agreement with the role of Ser[96] phosphorylation in regulating RhoA binding to RhoGDI and its activation. The fact that the effects of Ser[96] mutants were independent of p75[NTR] supports the specific role of the receptor in the upstream events leading to phosphorylation of Ser[34].

Together, these results support the notion that PKC-mediated and p75[NTR]-dependent phosphorylation of RhoGDI at Ser[34] triggers phosphorylation at Ser[96] leading to dissociation and activation of RhoA.

## The N-terminal domain of RhoGDI can interact on its own with p75[NTR] in a PKC-dependent manner

Previous work showed that the CTD of RhoGDI can interact on its own with the DD of p75[NTR]. Interestingly, mutations in DD residues that participate in binding significantly reduced the interaction but did not eliminate it (Lin et al, 2015). The location of Ser[34] within the RhoGDI NTD suggested that this domain may play a role in the interaction of RhoGDI with p75[NTR]. We generated a Flag-tagged RhoGDI construct lacking the NTD (RhoGDI-Δ59, Fig. 3A) and analyzed its interaction with p75[NTR] in transfected NIH3T3 cells after treatment with PMA or vehicle in comparison to wild-type RhoGDI (WT) and the S34A mutant. As reported above, we observed robust co-immunoprecipitation between WT RhoGDI and p75[NTR] which was greatly potentiated by PMA treatment (Fig. 3B). As expected, the S34A mutant interacted only weakly with p75[NTR] and was unaffected by PMA. RhoGDI-Δ59 was also detected in p75[NTR] immunoprecipitates, in agreement with the interaction of RhoGDI CTD with the DD of the receptor; however, its level was much lower than that of WT RhoGDI (Fig. 3B). PMA did not enhance interaction of RhoGDI-Δ59 with the receptor (Fig. 3B), indicating that the NTD of RhoGDI greatly potentiates the interaction, as well as making it sensitive to PKC activity. We then tested whether the RhoGDI NTD was on its own able to interact with p75[NTR]. To this end, we generated constructs carrying the 59 N-terminal residues of RhoGDI fused to Glutathione S-Transferase (GST), yielding a Flag-tagged fusion protein of about 35 kDa that could be easily detected in Western blotting studies (RhoGDI-59, Fig. 3D). RhoGDI-59 fusion proteins carrying either

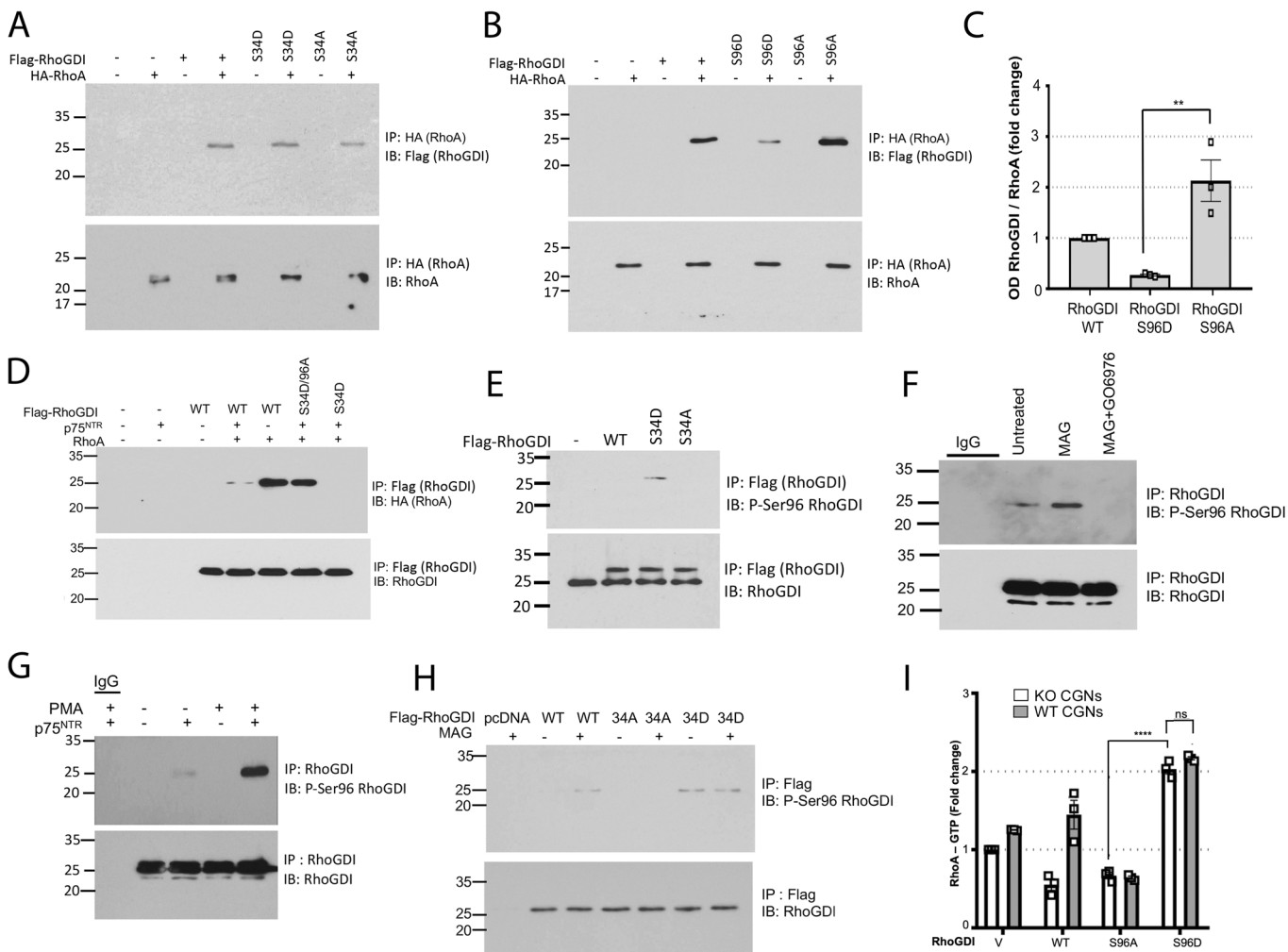

**Figure 2. Enhanced binding of RhoGDI to p75^NTR through phosphorylation of Ser^34 induces phosphorylation of Ser^96 leading to RhoA dissociation and activation.**

(A) Representative immunoblots of p75^NTR immunoprecipitation of protein from NIH3T3 cells cotransfected with p75NTR along with either RhoGDI wild type, S34D or S34A mutant. Western blots of whole cell lysates are shown in Appendix Fig. S2A. (B) Representative immunoblots of p75^NTR immunoprecipitation of protein from NIH3T3 cells cotransfected with p75^NTR along with either RhoGDI wild type, or Ser^96 single mutants of RhoGDI. Western blots of whole cell lysates are shown in Appendix Fig. S2B. (C) Quantification of RhoGDI in p75^NTR immunoprecipitates from NIH-3T3 cells shown in figure (B) after transfection with p75^NTR along with the indicated RhoGDI constructs. Values are shown as mean ± SEM from three independent experiments (**$p < 0.01$, one-way ANOVA followed by Tukey's multiple comparisons test). (D) Immunoblots of Flag-RhoGDI immunoprecipitation of protein from NIH3T3 cells transfected with either RhoGDI wild type, S34D/96A double mutant or S34D single mutants of RhoGDI. The RhoGDI immunoprecipitates were probed for bound HA-RhoA. Western blots of whole cell lysates are shown in Appendix Fig. S2C. (E) Representative immunoblots of Flag-RhoGDI immunoprecipitation of protein from P7 WT-CGNs neon transfected with constructs overexpressing either RhoGDI wild type, S34A or S34D mutants. Immunoblots were probed with antibodies to phospho-Ser^96 RhoGDI. Western blots of whole cell lysates are shown in Appendix Fig. S2D. (F) Representative immunoblot of endogenous RhoGDI immunoprecipitates from P7 CGNs. The CGNs were treated with either MAG (25 μM) or Gö6976 (1 μM) or a combination of both MAG and GO6976 for 30 min before harvesting. Immunoblots were probed with antibodies to phospho-Ser^96 RhoGDI. Western blots of whole cell lysates are shown in Appendix Fig. S2E. (G) Representative immunoblot of endogenous RhoGDI immunoprecipitates (or control IgG, as indicated) from P7 p75^NTR KO CGNs transduced with lentivirus expressing either empty vector (−) or p75^NTR (+). Cells were treated with PMA (10 ng/ml) for 30 min before harvesting. Immunoblots were probed with antibodies to phospho-Ser 96 RhoGDI. Western blots of whole cell lysates are shown in Appendix Fig. S2F. (H) Representative immunoblot of endogenous RhoGDI immunoprecipitates from P7 CGNs after neon transfection with WT, S34D or S34A RhoGDI. The CGNs were either left untreated or treated with MAG (25 μM) for 30 min before harvesting. Immunoblots were probed with antibodies to phospho-Ser^96 RhoGDI. Western blots of whole cell lysates are shown in Appendix Fig. S2G. (I) Quantification of active RhoA in cultured P7 KO or p75^NTR wild-type CGNs electroporated with constructs overexpressing either RhoGDI WT, S96A or S96D single mutants. Reading at 490 nm (mean ± SEM) from three independent experiments performed in triplicate is shown (****$p < 0.0001$, two-way ANOVA followed by Tukey's multiple comparison test). Source data are available online for this figure.

S34A or S34D mutations were also generated in a similar fashion (Fig. 3D), and their interaction with p75^NTR was assessed in transfected NIH3T3 cells after treatment with PMA or vehicle. Again, basal interaction between full-length WT RhoGDI and p75^NTR was robust and potentiated by PMA treatment (Fig. 3E).

Interestingly, RhoGDI-59 interacted quite strongly with p75^NTR, albeit only in the presence of PMA (Fig. 3E), indicating complete dependence on PKC activity. Mutant S34A RhoGDI-59 did not co-immunoprecipitate with p75^NTR, either in the presence or absence of PMA, and S34D interacted strongly with the receptor regardless

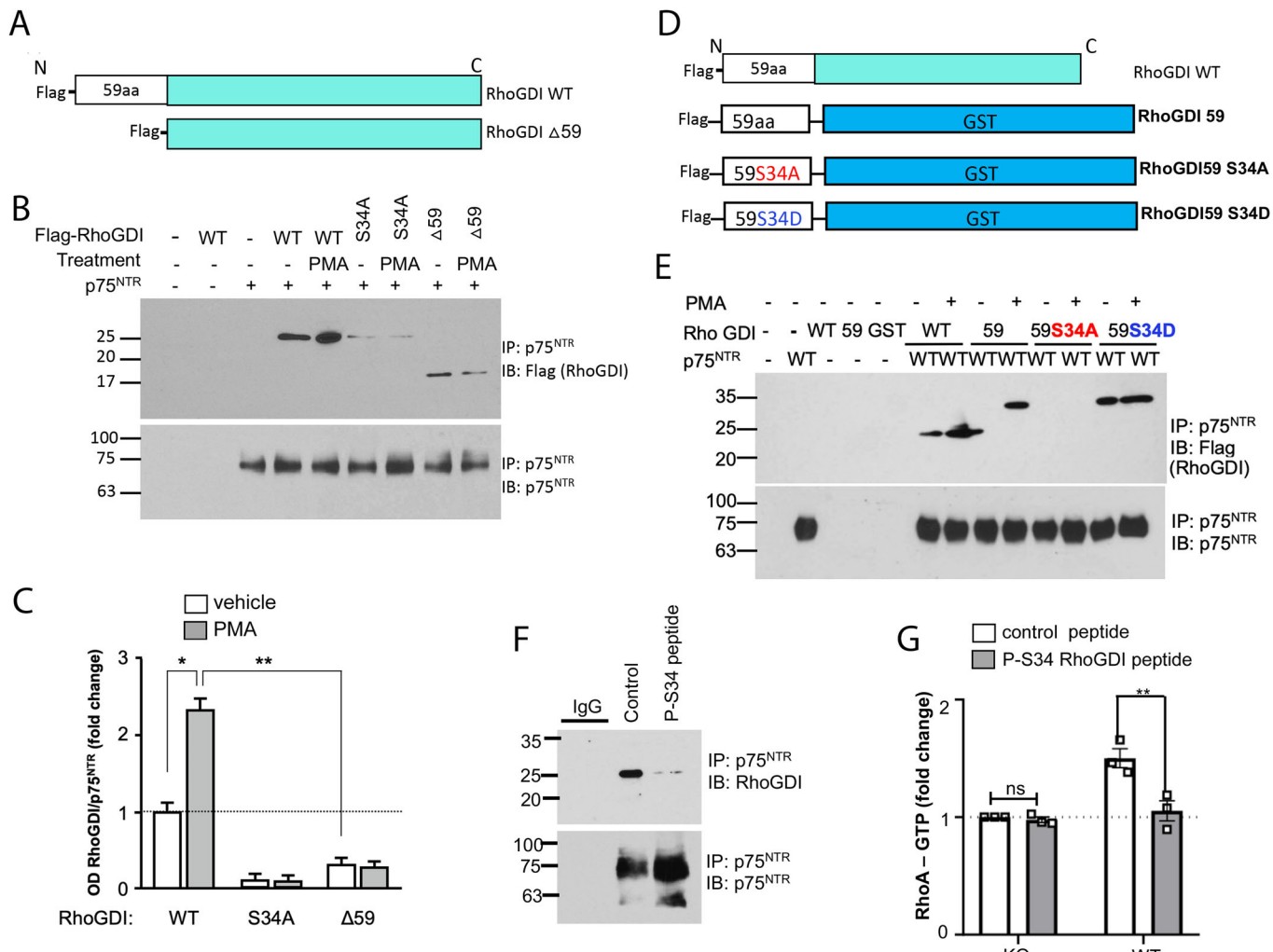

**Figure 3. The N-terminal domain of RhoGDI, including Ser[34], can interact on its own with p75[NTR] in a PKC-dependent manner.**

(A) Schematic representation of the deleted region of RhoGDI Δ59 with respect to the wild-type protein. (B) Representative immunoblots of p75[NTR] immunoprecipitation of protein from NIH3T3 cells cotransfected with p75[NTR] along with either RhoGDI wild type, or Δ59 after treatment with PMA or vehicle. Western blots of whole cell lysates are shown in Appendix Fig. S3A. (C) Quantification of RhoGDI in p75[NTR] immunoprecipitates shown in figure (B) after transfection with p75[NTR] along with the indicated RhoGDI constructs and treatment with PMA or vehicle. Values are shown as mean ± SEM from three independent experiments (*$p < 0.05$, **$p < 0.01$, two-way ANOVA followed by Tukey's multiple comparisons test). (D) Schematic representation of the mutated region of RhoGDI-59 with respect to the wild-type protein. (E) Representative immunoblots of p75[NTR] immunoprecipitation with either RhoGDI-59, or wild type. The higher molecular weight of RhoGDI-59 is due to the GST fusion protein. Western blots of whole cell lysates are shown in Appendix Fig. S3B. (F) Representative immunoblots of p75[NTR] immunoprecipitation of protein from cultured P7 p75[NTR] wild-type CGNs with lysates treated with 30 μM of control or RhoGDI phospho-Ser[34] peptide and then probed with antibodies to RhoGDI and p75[NTR]. Western blots of whole cell lysates are shown in Appendix Fig. S3C. (G) Quantification of active RhoA after treatment of lysates from cultured P7 KO or p75[NTR] wild-type lentivirus transduced KO CGNs with RhoGDI phospho S34 peptide. Mean ± SEM of reading at 490 nm from three independent experiments performed in triplicate is shown (**$p < 0.01$, two-way ANOVA followed by Tukey's multiple comparison test; ns, not significantly different). Source data are available online for this figure.

of PMA treatment (Fig. 3E). In primary CGN cultures, the interaction between endogenous p75[NTR] and RhoGDI could be reduced by addition of a synthetic 9-residue peptide derived from RhoGDI containing phosphorylated Ser[34] (Fig. 3F). The peptide also reduced basal levels of RhoA-GTP when introduced in wild type CGNs, but not in CGNs isolated from p75[NTR] knock-out mice (Fig. 3G). Together, these data show that the NTD of RhoGDI can on its own interact with p75[NTR] in a PKC-dependent manner. This interaction strengthens the recruitment of the RhoGDI/RhoA complex to the receptor and is required for the ability of p75[NTR] to enhance RhoA activity.

## RhoGDI NTD interacts with the JXT region of p75[NTR] through a salt bridge between RhoGDI phospho-Ser[34] and p75[NTR] Lys[303]

As the C-terminal region of RhoGDI interacts with the p75[NTR] DD, we considered the possibility that the RhoGDI NTD (RhoGDI-59) binds to the JXT domain of the receptor, which is N-terminal to the DD. Other downstream effectors of p75[NTR] are known to interact with this region, including TRAF6 (Khursigara et al, 1999) and NRIF (Casademunt et al, 1999). Inspection of the amino acid sequence of the JXT domain of p75[NTR] identified 4 candidate Lys

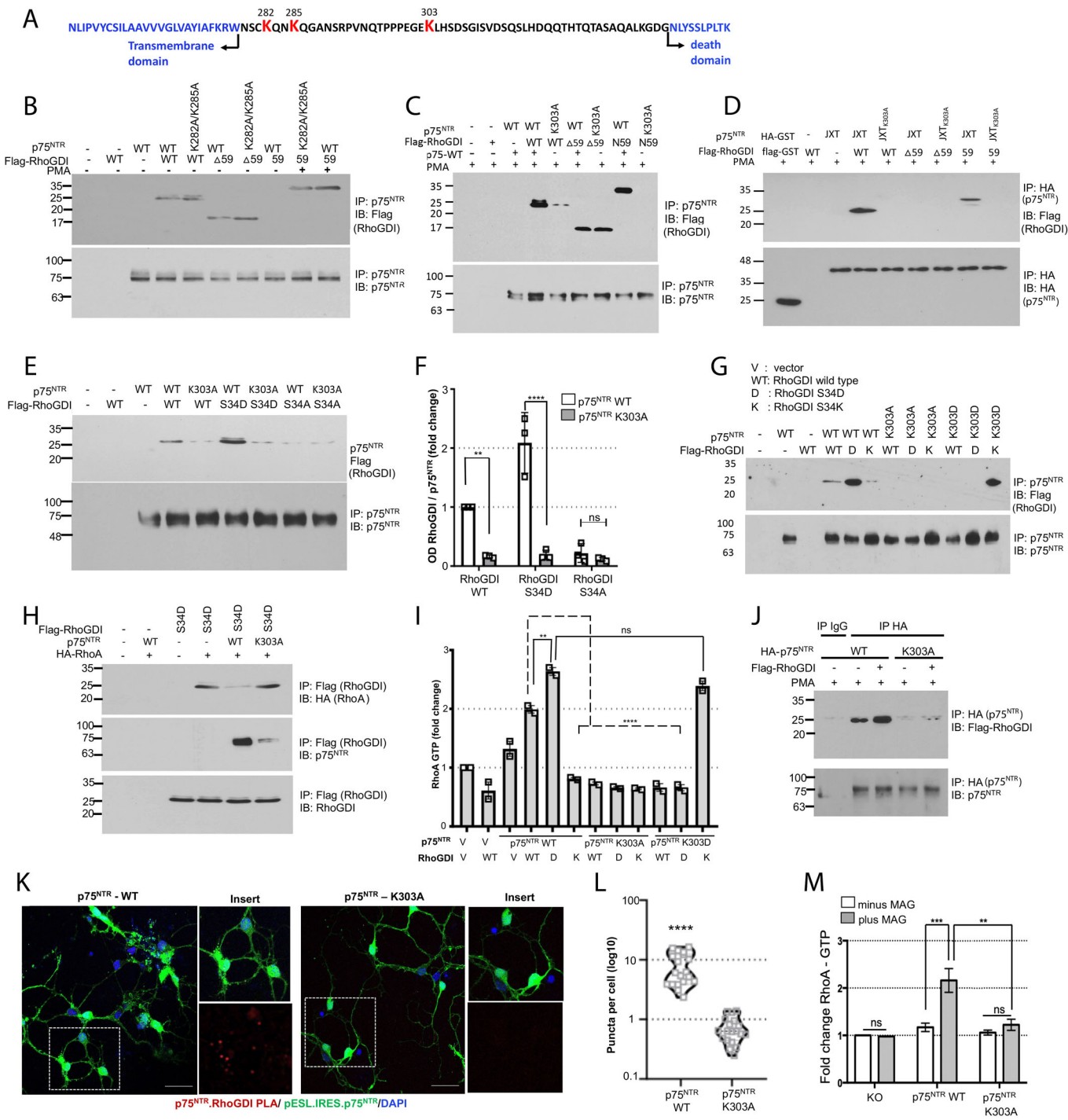

residues as potential sites for the formation of a salt bridge with phosphorylated Ser[34] in wild-type RhoGDI, or Asn[34] in the S34D RhoGDI mutant. We excluded the last Lys in this region as it was considered to be too close to the DD, leaving Lys[282], Lys[285] and Lys[303] as candidates (Fig. 4A). Co-immunoprecipitation assays between the p75[NTR] double mutant K282A/K285A and Flag-tagged wild type (WT) RhoGDI, RhoGDI-Δ59 or RhoGDI-59 constructs in transfected NIH3T3 cells (PMA was used in the latter case to induce p75[NTR] binding) demonstrated that neither Lys residue was

required for p75[NTR] interaction with RhoGDI (Fig. 4B). In contrast, binding of K303A p75[NTR] mutant to WT RhoGDI was greatly diminished (Fig. 4C), with the remaining binding provided by interaction between the p75[NTR] DD and RhoGDI CTD, as previously demonstrated (Lin 2015). Importantly, p75[NTR] lacking Lys[303] was correctly displayed at the plasma membrane (Appendix Fig. S4C), and the K303A mutation did not affect the binding of p75[NTR] to RhoGDI-Δ59, which lacks Ser[34] (Fig. 4C). No binding could be detected between K303A p75[NTR] and RhoGDI-59

**Figure 4. RhoGDI N-terminal domain interacts with the JXT region of p75$^{NTR}$ through a salt bridge between RhoGDI phospho-Ser$^{34}$ and p75$^{NTR}$ Lys$^{303}$.**

(A) Schematic representation of the location of K282, K285, and K303 in the juxtamembrane region of p75$^{NTR}$ with respect to the p75$^{NTR}$ death domain and the transmembrane domain. (B) Representative immunoblots of p75$^{NTR}$ K282A/K285A immunoprecipitation with either RhoGDI wild type or deletion mutants △59 or 59. PMA was used in the latter case to induce p75$^{NTR}$ binding. Western blots of whole cell lysates are shown in Appendix Fig. S4A. (C) Representative immunoblots of p75$^{NTR}$ immunoprecipitation of protein from NIH3T3 cells cotransfected with either p75$^{NTR}$ wild type or p75$^{NTR}$ K303A mutant along with either RhoGDI wild type, or deletion mutants of RhoGDI. Western blots of whole cell lysates are shown in Appendix Fig. S4B. (D) Representative immunoblots of p75$^{NTR}$ immunoprecipitation of protein from NIH3T3 cells cotransfected with the 'juxtamembrane alone' of p75NTR wild type or p75$^{NTR}$ K303A mutant along with either RhoGDI wild type, or deletion mutants of RhoGDI. Western blots of whole cell lysates are shown in Appendix Fig. S4D. (E) Representative immunoblots of p75$^{NTR}$ immunoprecipitation of protein from NIH3T3 cells cotransfected with the p75$^{NTR}$ wild type or p75$^{NTR}$ K303A mutant along with either RhoGDI wild type, or S34 mutants of RhoGDI. Western blots of whole cell lysates are shown in Appendix Fig. S4E. (F) Quantification of Flag-RhoGDI in p75$^{NTR}$ immunoprecipitates from NIH-3T3 cells after transfection with p75$^{NTR}$ along with the indicated RhoGDI constructs. Densitometric values are plotted as mean + SEM from 3 independent experiments and normalized to the WT control (**$p < 0.01$, ****$p < 0.0001$ by one-way ANOVA followed by Tukey's multiple comparisons test; ns, not significantly different). (G) Representative immunoblots of p75$^{NTR}$ WT, p75$^{NTR}$ K303A, or p75$^{NTR}$ K303D immunoprecipitation with RhoGDI wild type, RhoGDI S34D, and RhoGDI S34K. Western blots of whole cell lysates are shown in Appendix Fig. S4F. (H) Representative immunoblots of flag-RhoGDI immunoprecipitation of protein from NIH3T3 cells expressing either p75$^{NTR}$ WT or p75$^{NTR}$ K303A mutant. Western blots of whole cell lysates are shown in Appendix Fig. S4G. (I) Quantification of RhoA-GTP in NIH-3T3 cells after transfection with p75$^{NTR}$ WT or mutants along with the indicated RhoGDI constructs. Mean + SEM of fold change from 3 independent experiments is shown (**$p < 0.01$, ****$p < 0.0001$ one-way ANOVA followed by Tukey's multiple comparisons test; ns, not significantly different). (J) Representative immunoblots of HA-tagged p75$^{NTR}$ immunoprecipitation (or control IgG, as indicated) of protein from cultured p75$^{NTR}$ knock-out P7 CGNs transfected with plasmid constructs expressing either p75$^{NTR}$ wild type or K303A mutant and then probed with antibodies to RhoGDI and p75$^{NTR}$. Note the diminished interaction of RhoGDI with p75$^{NTR}$ K303A. Western blots of whole cell lysates are shown in Appendix Fig. S4H. (K) Micrographs of p75$^{NTR}$–RhoGDI PLA (red) in p75$^{NTR}$ KO CGNs transduced with lentivirus expressing p75$^{NTR}$ WT or p75$^{NTR}$ K303A mutant. The scale bars represent 20 μm. (L) Quantification of RhoGDI PLA puncta in p75$^{NTR}$ KO CGNs transduced with lentivirus expressing GFP-tagged p75$^{NTR}$ WT or p75$^{NTR}$ K303A mutant. Mean ± SEM of data from at least 200 neurons/condition from triplicate wells is shown (****$p < 0.0001$, paired t-test). (M) Quantification of active RhoA after treatment with or without MAG (25 μg/ml) in cultured P7 p75$^{NTR}$ KO CGNs transduced with lentivirus expressing either p75$^{NTR}$ wild type or p75$^{NTR}$ K303A mutant. Mean ± SEM of reading at 490 nm from 3 experiments performed in triplicate wells is shown (**$p < 0.01$, ***$p > 0.001$, two-way ANOVA followed by Tukey's multiple comparison test; ns, not significantly different). Source data are available online for this figure.

(fusion protein of RhoGDI N-terminus with GST) (Fig. 4C). In order to confirm the sufficiency of the p75$^{NTR}$ JXT domain for interaction with the RhoGDI N-terminus, we generated a (hemagglutinin-) HA-tagged fusion construct of the JXT domain and GST, as well as the corresponding K303A mutant, and tested their interaction with Flag-tagged RhoGDI constructs in transfected NIH3T3 cells. The wild-type JXT construct, but not the K303A mutant, was able to interact with both full-length (WT) RhoGDI and RhoGDI-59 (Fig. 4D). As expected, no binding between either JXT construct and RhoGDI-Δ59 was observed (Fig. 4D). The phosphorylation-mimetic RhoGDI mutant S34D potentiated binding to WT p75$^{NTR}$ but not to the K303A mutant, while S34A had no effect (Fig. 4E,F), in agreement with a polar interaction between P-Ser$^{34}$ and Lys$^{303}$. In order to confirm this possibility, we tested the effect of a charge switch between these two residues and generated S34K RhoGDI and K303D p75$^{NTR}$ mutant constructs. This experiment showed that the S34K/K303D pair interacted quite strongly, at levels comparable to the interaction of S34D RhoGDI with WT p75$^{NTR}$ (Fig. 4G). Other combinations did not yield an interaction. This result indicated that the Lys$^{303}$ side chain is dispensable *per se* as long as a salt bridge can be established between this position and residue 34 in the NTD of RhoGDI, also ruling out possible alternative roles for this residue via ubiquitination or methylation. Together the data supported the notion that a salt bridge between phosphorylated Ser$^{34}$ in RhoGDI and Lys$^{303}$ in p75$^{NTR}$ is critical for their interaction.

Our previous results indicated that S34D RhoGDI can interact with RhoA in the absence but not in the presence of p75$^{NTR}$, in agreement with the notion that strong binding of RhoGDI to p75$^{NTR}$ results in dissociation of RhoA from the complex (Fig. 2A,D). Unlike WT p75$^{NTR}$, however, the K303A mutant had no effect on the interaction between S34D RhoGDI and RhoA (Fig. 4H). This result indicates that, in the absence of p75$^{NTR}$, RhoGDI Ser$^{96}$ remains unphosphorylated and RhoA stays bound to RhoGDI, confirming that Ser$^{96}$ phosphorylation and RhoA release

from RhoGDI require recruitment of the latter to p75$^{NTR}$. We further tested the effects of combinations of Ser$^{34}$ and Lys$^{303}$ mutations on RhoA activation in transfected NIH3T3 cells. We found that the RhoA-GTP levels induced by different combinations of RhoGDI and p75$^{NTR}$ mutants (Fig. 4I) correlated with the strength of their interaction (Fig. 4G). In particular, K303D p75$^{NTR}$ was fully capable of stimulating RhoA activity in the presence of S34K, but not S34D, RhoGDI (Fig. 4I). These results reinforce the notion that p75$^{NTR}$ can switch RhoGDI from a RhoA inhibitor to a RhoA activator by facilitating its phosphorylation by PKC.

Lastly, we verified several of these findings in primary cultures of CGNs isolated from p75$^{NTR}$ KO mice in which HA-tagged forms of either p75$^{NTR}$ WT or K303A mutant were re-introduced by electroporation. In co-immunoprecipitation assays, we could show that PMA potentiates the interaction between WT RhoGDI and WT p75$^{NTR}$ but mutation of Lys$^{303}$ to Ala in p75$^{NTR}$ greatly reduced their interaction (Fig. 4J). A similar result was obtained in intact neurons using the Proximity Ligation Assay (PLA) and lentiviral transduction of GFP-tagged p75$^{NTR}$ constructs, in which WT p75$^{NTR}$ but not the K303A mutant was found to interact with endogenous RhoGDI (Fig. 4K,L). An earlier study suggested that MAG presented on the surface of CHO cells can activate the RhoA pathway independently of p75$^{NTR}$ in CGNs (Venkatesh et al, 2007). Using purified protein, however, we find that MAG is unable to activate RhoA in p75$^{NTR}$-KO neurons (Fig. 4M). Interestingly, MAG activity could be rescued by introducing WT p75$^{NTR}$ but not the K303A mutant (Fig. 4M), demonstrating that the ability of MAG to induce RhoA activity depends upon interaction between RhoGDI NTD and the JXT region of p75$^{NTR}$.

## NMR solution structure of RhoGDI NTD bound to p75$^{NTR}$ JXT domain

Earlier structural studies of the p75$^{NTR}$ intracellular domain found that the JXT region is unstructured and highly flexible in solution

(Liepinsh et al, 1997). Our finding of an interaction between this domain and the RhoGDI NTD suggested the possibility that their binding may impart structure and rigidity onto the p75[NTR] JXT domain. In order to investigate this further, two double-labeled samples with purities >95% of p75[NTR] JXT domain and RhoGDI NTD carrying the S34D mutation to allow binding (RhoGDI NTD[S34D]) were prepared for nuclear magnetic resonance (NMR) studies (Fig. EV1A). NMR titration of the p75[NTR] JXT domain to the [15]N-labeled RhoGDI NTD[S34D] showed that more than 50% of backbone cross peaks underwent noticeable chemical shift changes, indicating specific binding between the two domains (Fig. EV1B). Intermolecular NOEs were also detected from [13]C, [13]C -filtered 3D NOESY spectrum (Fig. EV1C). The solution structure of the complex was determined by multidimensional NMR spectroscopy (Table EV1). The ensemble of the ten lowest-energy structures of the p75[NTR] JXT:RhoGDI NTD[S34D] complex and a representative structure are depicted in Fig. 5A, B, respectively. In the complex,

RhoGDI NTD[S34D] folds into a three-helix small globular structure. A long loop of ~20 amino acid residues connects helices H1 and H2. The p75[NTR] JXT domain, which does not contain any secondary structure elements, wraps around the surface of RhoGDI NTD[S34D] (Fig. 5B). Analysis of the interface showed that both charge and hydrophobic interactions contribute to the formation of the complex structure (Fig. EV1D). In particular, the phosphorylation-mimicking Asp[34] on RhoGDI engages electrostatic interactions with side chains of Lys[303] and Asn[294] on the p75[NTR] JXT domain, stabilizing the local conformation of the long H1-H2 loop and facilitating the interaction of unstructured p75[NTR] JXT domain on the surface of the RhoGDI NTD (Fig. 5B). Previous structural studies showed that RhoGDI NTD can be induced to form three helices in a complex with RhoA (Fig. 5C) or Cdc42 (Fig. 5D) (Hoffman et al, 2000; Tnimov et al, 2012). Structural comparisons showed that the H1-H2 loop of RhoGDI NTD in the RhoGDI:R-hoA or RhoGDI:Cdc42 complexes remains in an extended

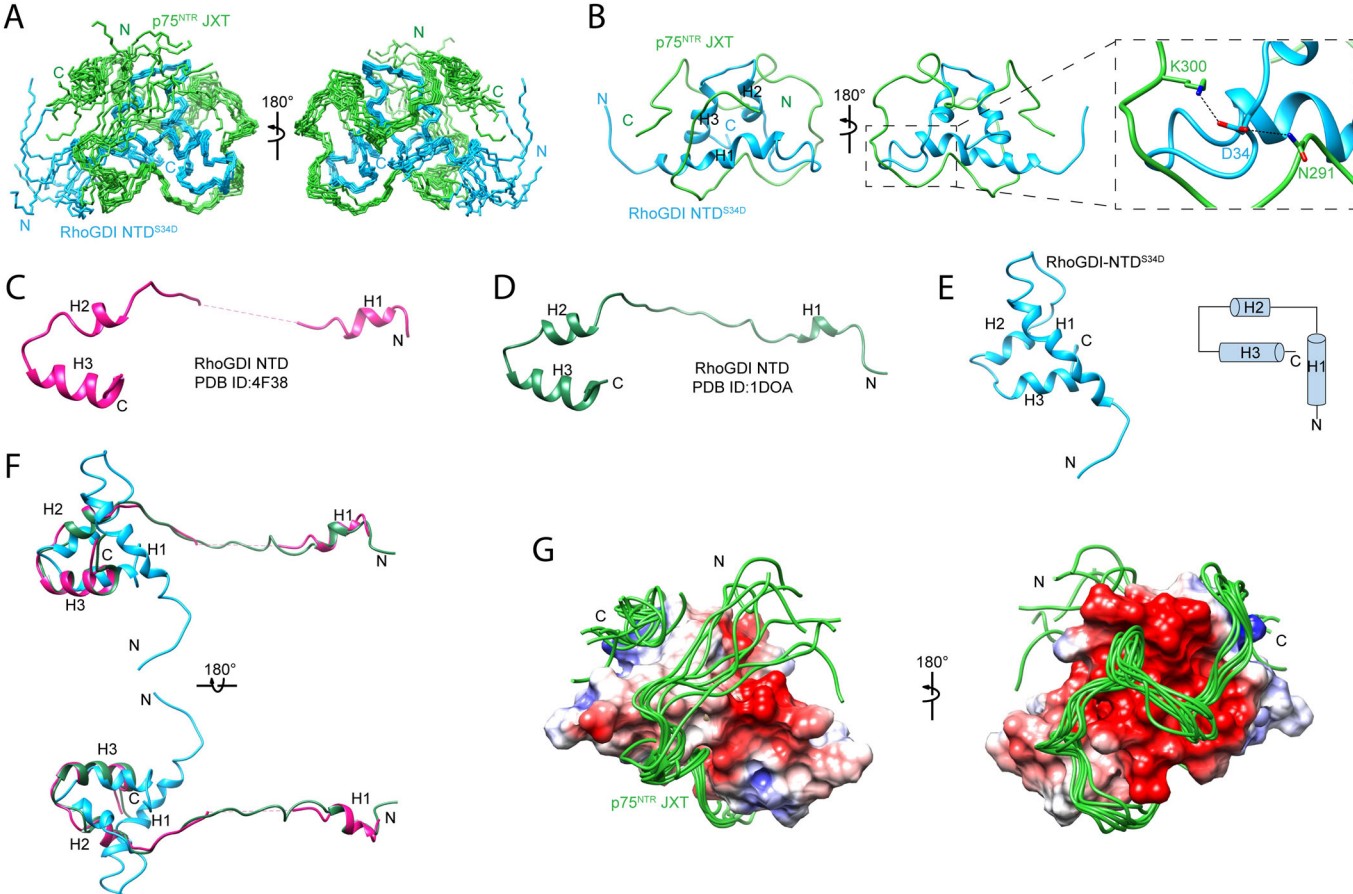

**Figure 5. Solution structure of the complex between the p75[NTR] JXT and RhoGDI NTD[S34D].**

(A) Superposition of backbone heavy atoms of the 10 lowest-energy structures of the human p75[NTR] JXT:RhoGDI NTD[S34D] complex. N- and C-termini are indicated. (B) Ribbon drawing of the lowest-energy conformer. Green, p75[NTR] JXT domain; Cyan, RhoGDI NTD[S34D]. N- and C-termini, as well as helices H1-H3 of RhoGDI NTD[S34D], are indicated. The expanded view shows detail of Asp[34] of RhoGDI NTD[S34D] interacting with side chains of Lys[300] and Asn[291] (highlighted as stick models). (C) Crystal structure of RhoGDI NTD from RhoGDI:RhoA complex (from PDB ID:4F38). (D) Crystal structure of RhoGDI NTD from RhoGDI:Cdc42 complex (from PDB ID: 1DOA). (E) NMR structure of RhoGDI NTD[S34D] from the p75[NTR] JXT:RhoGDI NTD[S34D] complex (left, from this study) and its helical topology (right). (F) Structural comparison between RhoGDI NTD[S34D] (cyan) from the p75[NTR] JXT:RhoGDI NTD[S34D] complex and RhoGDI NTD (magenta) from RhoGDI:RhoA complex (from PDB ID:4F38). (G) Wrapping of p75[NTR] JXT on the surface of RhoGDI NTD[S34D]. Disordered N-/C-termini of RhoGDI NTD[S34D] are not shown. Positive charge surface of RhoGDI NTD[S34D] is colored blue, negative red, and non-charged white. Source data are available online for this figure.

conformation, which is required for helix H1 to form hydrophobic interactions with the CTD of RhoGDI as well as the C-terminal tail of the GTPases. In the presence of the p75[NTR] JXT domain, helices H2 and H3 of phosphorylated RhoGDI NTD remain in a similar local conformation, but helix H1, which gains two extra turns, folds back to interact with both helices H2 and H3 (Fig. 5E,F). This new compact structure with a large negatively charged surface subsequently serves as a platform for the docking of flexible p75[NTR] JXT domain (Fig. 5G).

## PKC-mediated phosphorylation of RhoGDI at Ser[34] prevents RIP2 recruitment to p75[NTR] and NGF-mediated activation of the NF-kB pathway

Our previous work showed that the CTD of RhoGDI and RIP2 have a competitive, mutually exclusive interaction with the DD of p75[NTR] (Lin et al, 2015). Using purified proteins in solution, the affinity of RhoGDI CTD for the p75[NTR] DD was found to be more than 170-fold lower than that of the CARD domain of RIP2. In line with this, NGF engagement with p75[NTR] induces the recruitment of RIP2 and displacement of RhoGDI from the receptor. Our newly discovered interaction between the RhoGDI NTD and p75[NTR] JXT domain, raised the possibility that phosphorylation of RhoGDI at Ser[34] may substantially increase the affinity of RhoGDI for the receptor intracellular domain. Since the functional p75[NTR] is mostly present at the plasma membrane of neurons as a disulfide-linked dimer (Vilar et al, 2009), RhoGDI NTD and CTD may interact with p75[NTR] JXT and DD from either the same (cis) or different (trans) receptor protomers in the dimer (Fig. EV1E). Inspecting a structural model of the 2:2 complex between RhoGDI and the p75[NTR] DD (Lin et al, 2015), we noted that each RhoGDI NTD is in closer proximity to the N-terminus (N) of the DD that engages the CTD of the other RhoGDI molecule (arrows in Fig. EV1F), suggesting that each RhoGDI may interact in trans with the two protomers of the p75[NTR] dimer. In order to test this possibility, we used a complementation approach in a co-immunoprecipitation assay between RhoGDI and p75[NTR] in HEK-293 cells expressing WT p75[NTR], K303A mutant (which prevents binding of RhoGDI NTD to p75[NTR] JXT) or K346A/E349A (KEA) mutant (which prevents binding of RhoGDI CTD to p75NT DD) in various combinations (Fig. 6A). Although individually each p75[NTR] mutant bound very weakly or not at all to RhoGDI, their interaction could be rescued if the two mutants were expressed together (Fig. 6A), indicating that NTD and CTD of the same RhoGDI molecule can bind to JXT and DD, respectively, of different p75[NTR] molecules in the receptor dimer. Such configuration would be expected to considerably increase the binding strength between the two proteins and effectively lock the DD homodimer in a conformation that is no longer capable of recruiting RIP2. Thus, we decided to investigate how this may impact the relationship between RhoGDI and RIP2 with regards to p75[NTR] engagement and activation of their downstream pathways. As expected, and in contrast to MAG, we found that NGF treatment decreased the interaction between p75[NTR] and RhoGDI (both WT and S34A mutant) in transfected NIH3T3 cells but, interestingly, RhoGDI S34D was refractory to this effect and remained bound to the receptor (Fig. 6B). Moreover, while NGF reduced, and MAG increased, RhoA-GTP levels in CGNs expressing WT RhoGDI, neither ligand had any effect on neurons that received the S34D mutant, in which RhoA remained

maximally activated (Fig. 6C). While MAG induced phosphorylation of RhoGDI Ser[96] in CGNs, treatment of these neurons with NGF reduced baseline P-Ser[96] to levels below detection (Fig. 6D, see also 6E and F), indicating that NGF, in opposition to MAG, promotes de-phosphorylation of RhoGDI. Interestingly, the S34D RhoGDI mutant was largely resistant to Ser[96] de-phosphorylation in NGF-treated neurons (Fig. 6E), suggesting that persistent activation of PKC may overcome the effects of NGF on P-Ser[96]. Indeed, NGF was unable to induce P-Ser[96] de-phosphorylation in neurons that were concomitantly treated with MAG or PMA (Fig. 6F), suggesting a dominant relationship of MAG over NGF signaling.

These results prompted us to investigate the effects of the phosphorylation-mimetic S34D RhoGDI mutant on the competitive relationship between RhoGDI and RIP2 for binding to the p75[NTR] intracellular domain. Using epitope-tagged constructs in transfected NIH3T3 cells, we found that, unlike WT RhoGDI, S34D RhoGDI was resistant to displacement from p75[NTR] by increasing doses of RIP2 (Fig. 6G). At the same time, RIP2 was displaced from p75[NTR] by increasing doses of S34D RhoGDI (Fig. 6G). However, S34D RhoGDI was unable to displace RIP2 if the K303A mutation was introduced in p75[NTR] (Fig. 6H). To assess NF-kB activation, we used an assay in which colorimetric readout at 450 nm measures activated NF-kB p65 subunit on whole cell extracts bound to oligonucleotide containing p65 consensus DNA binding sites immobilized to the 96-well plate. In CGNs, NGF treatment enhanced NF-kB activation in neurons electroporated with vector, WT RhoGDI or S34A RhoGDI but not in neurons that received the S34D RhoGDI mutant (Fig. 6I), indicating that strong binding of RhoGDI to p75[NTR] prevents NGF-mediated RIP2 recruitment and activation of NF-kB in primary neurons. In agreement with the effects of the S34D RhoGDI mutant, we found that PMA treatment of CGNs completely disengaged RIP2 from the receptor, even in the presence of NGF (Fig. 6J). This effect was mediated by RhoGDI, as RIP2 binding to p75[NTR] could be restored by siRNA-mediated knock-down of RhoGDI (Fig. 6K). Likewise, treatment of CGN neurons with MAG reduced RIP2 binding to p75[NTR] even in the presence of NGF (Fig. 6L,M). When applied individually, NGF reduced and MAG enhanced RhoA activation (Fig. 6N), while the opposite was the case for NF-kB activation (Fig. 6O). However, when applied simultaneously, RhoA activation was enhanced and NF-kB activation was suppressed (Fig. 6N,O, respectively), indicating that MAG prevails over NGF if encountered concomitantly by neurons responsive to both factors. On the other hand, MAG was unable to suppress NF-kB activation by NGF in p75[NTR] KO neurons that were reconstituted with a lentivirus expressing the K303A mutant (Fig. 6P). Together these results reveal an unexpected tug-of-war between MAG and NGF in the activation of p75[NTR] downstream pathways resulting from an underlying competition, regulated by PKC, between RhoGDI and RIP2 for access to the receptor intracellular domain.

## RhoGDI/p75[NTR] interaction through phospho-Ser[34]/Lys[303] bonding is critical for MAG effects axon growth and neuron survival

The functional relevance of our newly discovered interaction between RhoGDI and p75[NTR] through phospho-Ser[34]/Lys[303] bonding

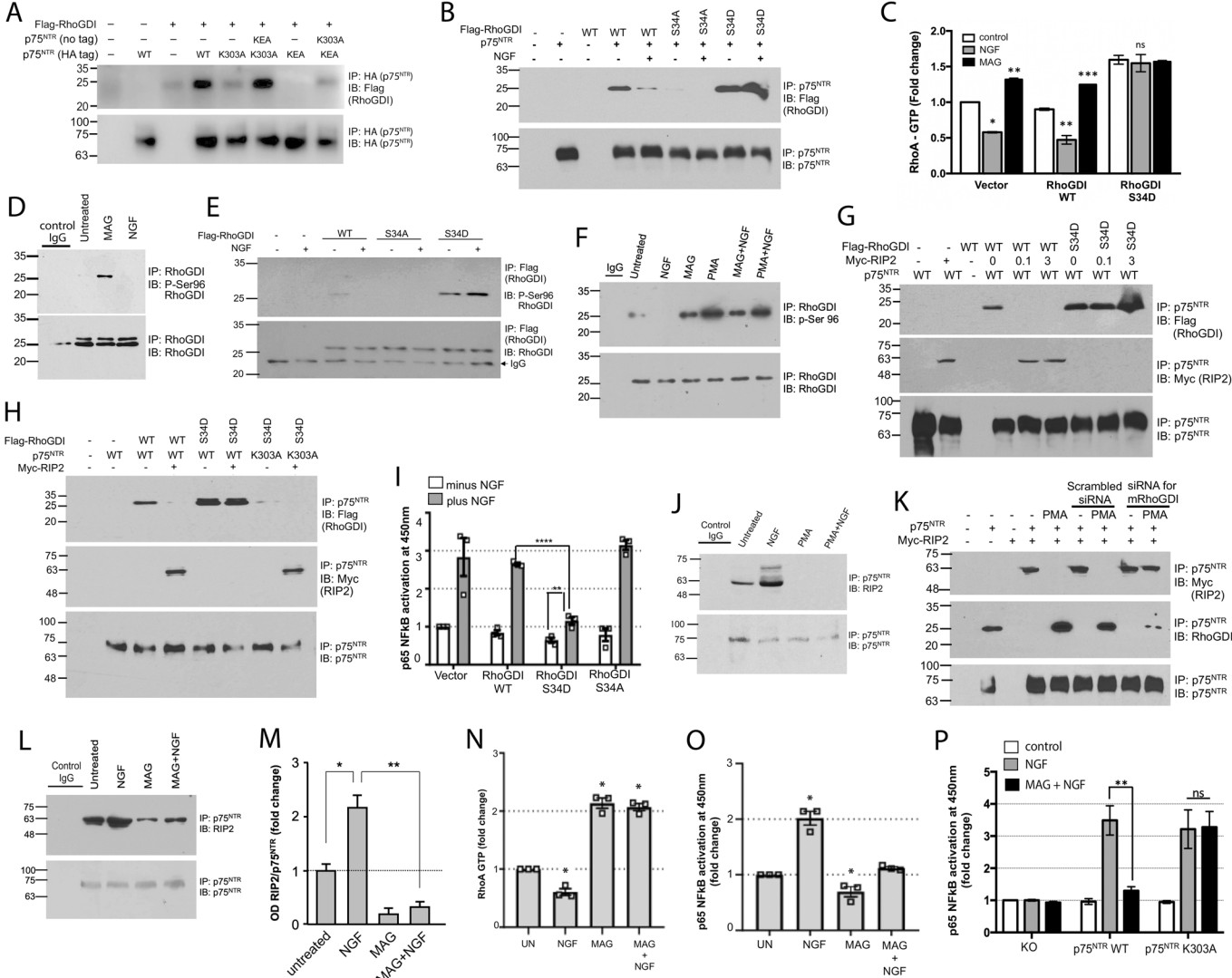

was investigated in primary neuronal cultures of CGNs. We used gain- and loss-of-function strategies based on lentiviral vectors directing expression of either RhoGDI S34D or p75[NTR] K303A mutants, respectively, and assessed growth cone collapse and neurite outgrowth, two key biological activities attributed to the MAG→p75[NTR]→RhoA pathway in these neurons (Wang et al, 2002a; Wong et al, 2002; Park et al, 2010; Yamashita et al, 2002; Yamashita and Tohyama, 2003). Expression of RhoGDI S34D induced robust (≈90%) growth cone collapse in wild-type CGNs but was inactive in neurons isolated from p75[NTR] KO mice (Fig. 7A). MAG treatment replicated this activity (≈70% growth cone collapse) in p75[NTR] knock-out neurons reconstituted with wild type p75[NTR], but was inactive in neurons that received the K303A mutant (Fig. 7B). In agreement with these results, p75[NTR]-expressing neurons that received the RhoGDI S34D construct showed reduced neurite outgrowth (≈50% length reduction), while p75[NTR] knock-out neurons remained unaffected (Fig. 7C). In the presence of MAG, neurons expressing wild type p75 showed reduced neurite outgrowth (≈30% length reduction), but neurons expressing the K303A mutant were resistant to this effect (Fig. 7D).

These results indicate that interaction of RhoGDI with p75[NTR] through phospho-Ser[34]/Lys[303] bonding is critical for the effects of MAG, and likely other myelin-derived molecules, on growth cone stability and neurite growth.

It has been recently established that MAG can also induce apoptotic cell death in CGNs (Fernández-Suárez et al, 2019). We assessed apoptotic cell death in cultured CGNs through immunocytochemistry for cleaved caspase-3. It should be noted that, in our culture conditions, MAG induced activation of caspase-3 in a much smaller proportion of neurons (<10%) and in a longer time-scale than its effects on growth cone collapse (Fig. 7E,F). The effects of MAG on cell death were mimicked by expression of the RhoGDI S34D mutant (Fig. 7F,G). In this case however, MAG treatment did not affect the level of activated caspase-3 beyond that achieved by the S34D construct (Fig. 7F,G), suggesting maximal activation of this pathway by the RhoGDI mutant. In agreement with these results, only neurons expressing wild-type p75[NTR] were affected by MAG (Fig. 7H,I). Neurons lacking p75[NTR] (vector) or expressing the K303A mutant were resistant to MAG-induced cell death (Fig. 7H,I). Together these data demonstrate the essential role of

**Figure 6. PKC-mediated phosphorylation of RhoGDI at Ser$^{34}$ prevents RIP2 recruitment to p75$^{NTR}$ and NGF-mediated activation of the NF-kB pathway.**

(A) Immunoblots of p75$^{NTR}$ immunoprecipitation of protein from HEK293 cells transfected with Flag-RhoGDI, and HA-tagged or untagged constructs of wild type (WT) p75NTR, K303A mutant or K346A/E349A (KEA) mutant, as indicated, probed with antibodies to Flag (upper) and re-reprobed with anti-HA antibodies (lower). Western blots of whole cell lysates are shown in Appendix Fig. S5A. (B) Immunoblots of p75$^{NTR}$ immunoprecipitation of protein from NIH3T3 cells treated in the presence or absence of NGF (100 ng/ml) and then probed with antibodies to Flag (RhoGDI) and p75$^{NTR}$. Note the stable interaction of RhoGDI S34D mutant with p75NTR in the presence of NGF. Western blots of whole cell lysates are shown in Appendix Fig. S5B. (C) Quantification of active RhoA after treatment with NGF (100 ng/ml) or MAG (25 µg/ml) in cultured P7 p75$^{NTR}$ WT CGNs transduced with lentivirus expressing either RhoGDI wild type or S34D mutant. Mean ± SEM of reading at 490 nm from 3 experiments performed in triplicate wells is shown (*$p > 0.01$, **$p > 0.002$, ****$p > 0.0001$, two-way ANOVA followed by Tukey's multiple comparison test; ns, not significantly different). (D) Representative immunoblot of endogenous RhoGDI immunoprecipitates from P7 CGNs. The CGNs were treated with MAG (25 µg/ml) or NGF (100 ng/ml) for 30 min. Immunoblots were probed with antibodies to phospho-Ser 96 RhoGDI. Western blots of whole cell lysates are shown in Appendix Fig. S5C. (E) Representative immunoblots of Flag-RhoGDI immunoprecipitation of protein from WT-CGNs treated in the presence or absence of NGF (100 ng/ml) and then probed with antibodies to antibodies to phospho-Ser$^{96}$ RhoGDI. Note the reduction in phospho-Ser$^{96}$ RhoGDI levels in the RhoGDI-WT immunoprecipitates in the presence of NGF. Western blots of whole cell lysates are shown in Appendix Fig. S5D. (F) Immunoblots of Flag-RhoGDI immunoprecipitation of protein from WT-CGNs treated in the presence of NGF, MAG, or PMA and then probed with antibodies to antibodies to phospho-Ser 96 RhoGDI. Note that co-treatment of CGNs with MAG or PMA prevents the NGF-mediated reduction in phospho-Ser 96 RhoGDI levels in the RhoGDI-WT immunoprecipitates. Western blots of whole cell lysates are shown in Appendix Fig. S5E. (G) Representative immunoblots of a competition between RIP2 and RhoGDI S34D mutant for binding to p75$^{NTR}$. p75$^{NTR}$ was immunoprecipitated from NIH3T3 cells transfected with p75$^{NTR}$, RIP2 and either RhoGDI wild type or RhoGDI S34D mutant and then probed with antibodies to RhoGDI, RIP2 and p75$^{NTR}$. Note the diminished interaction of RIP2 with p75NTR in the presence of RhoGDI S34D mutant. Western blots of whole cell lysates are shown in Appendix Fig. S5F. (H) Representative RhoGDI immunoblots of p75$^{NTR}$ immunoprecipitates from NIH3T3 cells cotransfected with either p75$^{NTR}$ wild type or p75$^{NTR}$ K303A mutant along with Myc-RIP2 and RhoGDI. Note that p75$^{NTR}$-RIP2 binding is rescued in the presence of RhoGDI S34D in NIH3T3 cells expressing p75$^{NTR}$-K303A mutant. Western blots of whole cell lysates are shown in Appendix Fig. S5G. (I) Quantification of active p65 NF-kB in cultured P7 wild-type CGNs electroporated with RhoGDI wild type, S34D or S34A mutant in the presence or absence of NGF. Mean ± SEM of reading at 450 nm from 3 independent experiments performed in triplicate is shown (**$p < 0.01$, ****$p < 0.0001$, two-way ANOVA followed by Tukey's multiple comparison test; ns, not significantly different). (J) Representative RIP2 immunoblots of p75$^{NTR}$ immunoprecipitates from cultured P7 p75$^{NTR}$ wild-type CGNs treated with either NGF or PMA or a combination of both for 30 min and then probed with antibodies to RIP2 and p75$^{NTR}$. Note the diminished interaction of RIP2 with p75$^{NTR}$ in the presence of PMA. Western blots of whole cell lysates are shown in Appendix Fig. S5H. (K) RIP2 immunoblots of p75$^{NTR}$ immunoprecipitates from NIH3T3 cells overexpressing p75$^{NTR}$ and RIP2. The cells were treated with either scrambled siRNA or RhoGDI siRNA in the presence or absence of PMA and then probed with antibodies to RIP2, p75$^{NTR}$, and endogenous RhoGDI. Western blots of whole cell lysates are shown in Appendix Fig. S5I. (L) RIP2 immunoblots of p75$^{NTR}$ immunoprecipitates (or control IgG, as indicated) from p75$^{NTR}$ WT CGNs. The cells were treated with either NGF, MAG or a combination of both NGF and MAG for 30 min before harvesting for immunoprecipitation and then probed with antibodies to endogenous RIP2 and p75$^{NTR}$. Western blots of whole cell lysates are shown in Appendix Fig. S5J. (M) Quantification of RIP2 in p75$^{NTR}$ immunoprecipitates shown in figure (K) after treatment with NGF, MAG or their combination as indicated. Values are shown as mean ± SEM from three independent experiments (*$p < 0.05$, **$p < 0.01$, two-way ANOVA followed by Tukey's multiple comparisons test). (N) Quantification of active RhoA in cultured P7 p75$^{NTR}$ wild-type CGNs treated with NGF, MAG, or MAG and NGF in combination for 30 min, untreated. Values are shown as mean ± SEM from three independent experiments performed in triplicate (*$p < 0.05$ vs. UN by one-way ANOVA followed by Tukey's multiple comparisons test). (O) Quantification of active p65 NF-kB in cultured P7 p75$^{NTR}$ wild-type CGNs treated with either NGF (100 ng/ml), MAG (25 µg/ml) or a combination of both NGF and MAG for 30 min. UN, untreated. Values are shown as mean ± SEM from three independent experiments performed in triplicate (*$p < 0.05$ vs. UN by one-way ANOVA followed by Tukey's multiple comparisons test). (P) Quantification of active p65 NF-kB in cultured P7 p75$^{NTR}$ knock-out (KO) CGNs transduced with lentivirus expressing either p75$^{NTR}$ wild type or p75$^{NTR}$ K303A mutant. After 2 days in vitro post transduction, the CGNs were treated with either NGF (100 ng/ml), MAG (25 µg/ml) or a combination of both NGF and MAG for 30 min. Reading at 450 nm (mean ± SEM) from 3 experiments performed in triplicates is shown (**$p < 0.01$ vs. KO, two-way ANOVA followed by Tukey's multiple comparison test; ns, not significantly different). Source data are available online for this figure.

phospho-Ser$^{34}$/Lys$^{303}$ bonding between RhoGDI and p75$^{NTR}$ for all the known MAG activities mediated by p75$^{NTR}$.

## Discussion

Decades after the discovery of p75$^{NTR}$, a holistic understanding of the logic underlying its function remains elusive. Over the years, multiple proteins were identified as intracellular interactors of the receptor, linking to a wide variety of signaling pathways and biological activities. p75$^{NTR}$ would appear to be the proverbial Swiss Army knife of cellular signaling (Chao, 2003). It can contribute to neuron survival (e.g., (Meier et al, 2019; Khursigara et al, 2001)), but also induce apoptotic cell death (e.g., (Volosin et al, 2006; Lee et al, 2001)). It can promote axonal growth (e.g., (Bentley and Lee, 2000)), but also induce growth cone collapse (e.g., (Deinhardt et al, 2011)) and inhibit nerve regeneration (e.g., (Fujita et al, 2011)). It can promote cell proliferation (e.g., (Truzzi et al, 2008)), but also induce cell cycle withdrawal (e.g., (Cragnolini et al, 2011; Zanin et al, 2016)). Reconciling these seemingly contradictory pathways and activities will require a more integrated understanding of p75$^{NTR}$ function under different conditions and situations. Front and center is the issue of cellular context. Receptors are switches,

and just like any switch, they can turn signals on or off depending on how they are wired. For instance, p75$^{NTR}$ can induce NF-kB signaling in CGNs by coupling to RIP2, but not in hippocampal neurons, which do not express RIP2 at sufficient levels (Vicario et al, 2015). Even within the same cell type, p75$^{NTR}$ can induce different, and sometimes opposed, activities depending on the nature of the incoming ligand. In CGNs, for example, pro-neurotrophins induce cell death, while mature NGF promotes cell survival, as the latter but not the former can enhance the activity of NF-kB (Kisiswa et al, 2018). In vivo, p75$^{NTR}$ is likely to encounter several ligands simultaneously, but it remains unclear how the receptor integrates such signals to generate a coherent physiological response. A concept that has emerged from recent work centers on the competitive, i.e., mutually exclusive, nature of the interactions of different downstream effectors with the p75$^{NTR}$ intracellular domain (Lin et al, 2015; Kisiswa et al, 2018). In the present study, we have elucidated the mechanism by which MAG induces RhoA activity through p75$^{NTR}$ and PKC, and demonstrated that competition between RIP2 and RhoGDI for receptor binding can go either way, depending on whether the receptor is exposed to NGF or MAG, respectively. Intriguingly, when both ligands are present, our findings indicate that MAG predominates over NGF and RhoGDI outcompetes RIP2.

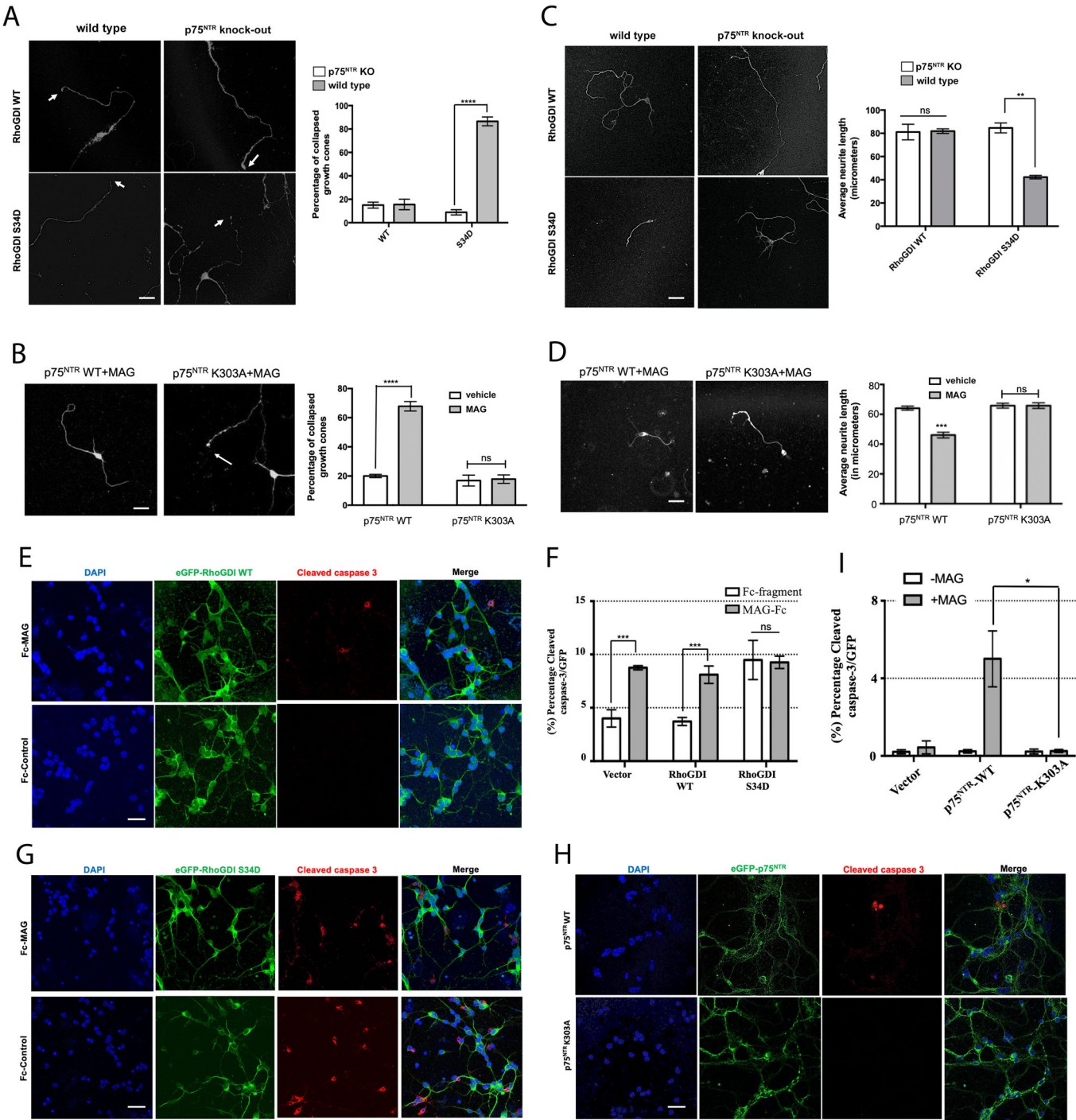

RhoGDI can interact through its CTD in a constitutive manner with the p75[NTR] DD under baseline conditions and in the absence of external stimuli. However, the affinity of such interaction is low, accounts for only a modest stimulation of RhoA activity and can be outcompeted with relative ease by other downstream effectors of the receptor, such as RIP2 (Lin et al, 2015). A constitutive interaction between RhoGDI and p75[NTR] has also left unexplained how MAG treatment is able to increase binding of RhoGDI to the receptor and RhoA activity. Our finding of the critical role played by PKC-mediated phosphorylation of RhoGDI at Ser[34] for its

binding to the JXT of p75[NTR] complements previous studies by us and others, and illuminates the mechanism by which MAG and likely other myelin-derived ligands are able to increase the binding of RhoGDI to the receptor as well as stimulate RhoA activity. We hypothesize that the CTD/DD interaction helps to initially recruit RhoGDI to p75[NTR] through a contact that is weaker than that of RIP2 and which can be displaced upon NGF binding to the receptor. On the other hand, MAG and other ligands that enhance PKC activity can cement RhoGDI binding to p75[NTR] by inducing the NTD/JXT interaction. Our observation that CTD/DD and

◀ **Figure 7. RhoGDI/p75NTR interaction through phospho-Ser34/Lys303 bonding is critical for MAG effects axon growth and neuron survival.**

(A) Representative micrographs of GFP fluorescence in cultured p75NTR WT or KO CGNs transduced with lentivirus expressing GFP and either RhoGDI wild type, or S34D mutant. White arrows denote growth cones. The graph shows percentage of growth cones (mean ± SEM) based on three experiments each performed in triplicate wells (****$p < 0.0001$, two-way ANOVA followed by Tukey's multiple comparison test). Scale bar, 10 μm. (B) Representative micrographs of GFP fluorescence in cultured p75NTR KO CGNs transduced with lentivirus expressing GFP and either WT or K303A p75NTR. White arrows denote growth cones. Axonal integrity was verified by counterstaining for Tuj1 (Appendix Fig. S6). The graph shows percentage of growth cones (mean ± SEM) 30 min post addition of MAG (25 μg/ml) based on three experiments each performed in triplicate wells (****$p < 0.0001$, two-way ANOVA followed by Tukey's multiple comparison test; ns, not significantly different). Scale bar, 10 μm. (C) The graph shows average neurite length of CGNs expressing RhoGDI wild type (WT) or S34D mutant in the presence or absence of p75NTR. p75NTR WT or KO CGNs were transduced with lentivirus expressing wither RhoGDI wild type, or S34D mutant. Result is shown as mean ± SEM based on three experiments performed in triplicate (**$p < 0.01$, two-way ANOVA followed by Tukey's multiple comparison test; ns, not significantly different). Scale bar, 10 μm. (D) The graph shows average neurite length of CGNs expressing p75NTR WT or p75NTR K303A mutant in the presence of MAG (25 μg/ml) overnight. Result is shown as mean ± SEM based on three experiments performed in triplicate (***$p < 0.001$, two-way ANOVA followed by Tukey's multiple comparison test; ns, not significantly different). Scale bar, 10 μm. (E) Representative micrographs of cultured CGNs from p75NTR WT mice transduced with lentivirus expressing eGFP-tagged RhoGDI-WT followed by treatment with Fc fragment (control) or MAG-fusion protein (MAG-Fc) for overnight. The cells were labeled with antibodies to cleaved caspase 3 (red) and counterstained with DAPI. Scale bar, 10 μm. (F) Quantification of experiment (E and G) to obtain percentage cells positive for cleaved caspase 3 relative to DAPI. Result is shown as mean ± SEM based on three experiments performed in triplicate (***$p < 0.001$, two-way ANOVA followed by Tukey's multiple comparison test; ns, not significantly different). (G) Representative micrographs of cultured CGNs from p75NTR-WT mice transduced with lentivirus expressing eGFP-tagged RhoGDI S34D mutant followed by treatment with Fc fragment (control) or MAG-fusion protein (MAG-Fc) overnight. The cells were labeled with antibodies to cleaved caspase 3 and counterstained with DAPI. Scale bar, 10 μm. (H) Representative micrographs of cultured CGNs from p75NTR KO mice transduced with lentivirus expressing eGFP-tagged p75NTR WT or K303A mutant followed by treatment with MAG-fusion protein (MAG-Fc) for overnight. The cells were labeled with antibodies to cleaved caspase 3 (red) and counterstained with DAPI. Scale bar, 10 μm. (I) Quantification of experiment (H) to obtain percentage cells positive for cleaved caspase 3 relative to DAPI. Result is shown as mean ± SEM based on three experiments performed in triplicate (*$p < 0.05$, two-way ANOVA followed by Tukey's multiple comparison test). Source data are available online for this figure.

NTD/JXT interactions can occur in "trans" across two p75NTR molecules would effectively lock RhoGDI onto the receptor intracellular domain, thereby shielding the RhoGDI/p75NTR complex from the effects of NGF and RIP2 (Fig. 8).

Previous work had alternatively implicated phosphorylation of either Ser34 or Ser96 in the mechanism by which PKC stimulates the release of RhoA from RhoGDI allowing its activation at the plasma membrane (Dovas et al, 2010; Knezevic et al, 2007; Sabbatini and Williams, 2013). Our present findings firmly establish Ser96 in this role, ruling out the participation of Ser34, which would appear to be solely involved in interaction with p75NTR JXT domain. In this regard, our finding that Ser96 phosphorylation requires prior phosphorylation at Ser34 may in part explain the discrepancy among previous studies. This result also suggests that robust recruitment of RhoGDI to p75NTR is required for Ser96 phosphorylation and RhoA release. It is also interesting to note that phosphorylation of Ser34 significantly increases the negative charge of the RhoGDI NTD, causing strong electrostatic repulsion of nearby Glu121 from the RhoGDI CTD (Fig. EV1G), possibly leading to dissociation of the RhoGDI NTD from the RhoGDI CTD and subsequently from RhoA. Previous structural and biochemical studies have indicated that RhoGDI CTD binding to the p75NTR DD induces conformational changes that facilitate the release of RhoA from RhoGDI (Lin et al, 2015). Our present results suggest that dissociation of RhoGDI NTD and CTD upon Ser34 phosphorylation is an additional mechanism contributing to the release of RhoA to allow its activation at the plasma membrane. Upon phosphorylation at Ser34, RhoGDI could outcompete RIP2 for binding to p75NTR, indicating a significant increase in affinity, explaining how MAG can overcome NGF when both ligands are presented simultaneously.

The JXT domain of p75NTR is a 60 residue long and highly flexible region linking the DD to the transmembrane domain. Earlier NMR studies have shown it to lack both secondary and tertiary structures in solution (Liepinsh et al, 1997). This has presented a problem for understanding how conformational changes induced by ligand binding to the extracellular domain

can be transmitted to the intracellular region of the receptor, specially to the DD, as a highly flexible and unstructured JXT domain would effectively prevent any movement occurring in the N-terminal regions of the molecule from reaching the C-terminally located DD. Current models of ligand-mediated p75NTR activation and signaling crucially depend on the feasibility of such transfer of conformational information, for which the disulfide link made by Cys259 in the transmembrane domain is also essential (Vilar et al, 2009). Our discovery of the interaction between p75NTR JXT domain and the N-terminal region of RhoGDI, both nearly identical in size, offers the possibility that such contact may confer structural rigidity to the JXT domain. Indeed, as shown by our NMR structure analysis, the p75NTR JXT domain, although still lacking secondary structure, is shown to wrap around the RhoGDI NTD making multiple contacts with it in the complex. This suggests that, in the cell cytoplasm, the p75NTR JXT domain may in fact be rigid owing to its interaction with downstream targets, such as RhoGDI NTD. In this way, the JXT domain would be capable of transmitting conformation changes occurring in the extracellular and transmembrane domains to the DD in the C-terminal region of the receptor.

MAG promotes strong binding of RhoGDI to p75NTR by inducing its phosphorylation at Ser34, leading to phosphorylation of Ser96 and subsequent release and activation of RhoA. This series of events require PKC activity, but it remains unclear whether or how p75NTR mediates PKC activation in response to MAG. In CGNs, p75NTR expression enhances the phosphorylation of several PKC substrates (Fig. EV2) and MAG-induced phosphorylation of PKC substrates is markedly reduced in p75NTR KO neurons (Fig. EV2, see also (Sivasankaran et al, 2004)), suggesting the involvement of p75NTR in PKC activation. Interestingly, earlier experiments in transfected HEK cells indicated that MAG can elevate intracellular $Ca^{2+}$ levels when p75NTR is introduced into the cells (Wong et al, 2002). At present, however, none of the known downstream effectors of p75NTR offers a direct link to increased $Ca^{2+}$ or diacylglycerol (DAG) levels, the two main activation signals of PKC. Activated PKC has been shown to translocate to the

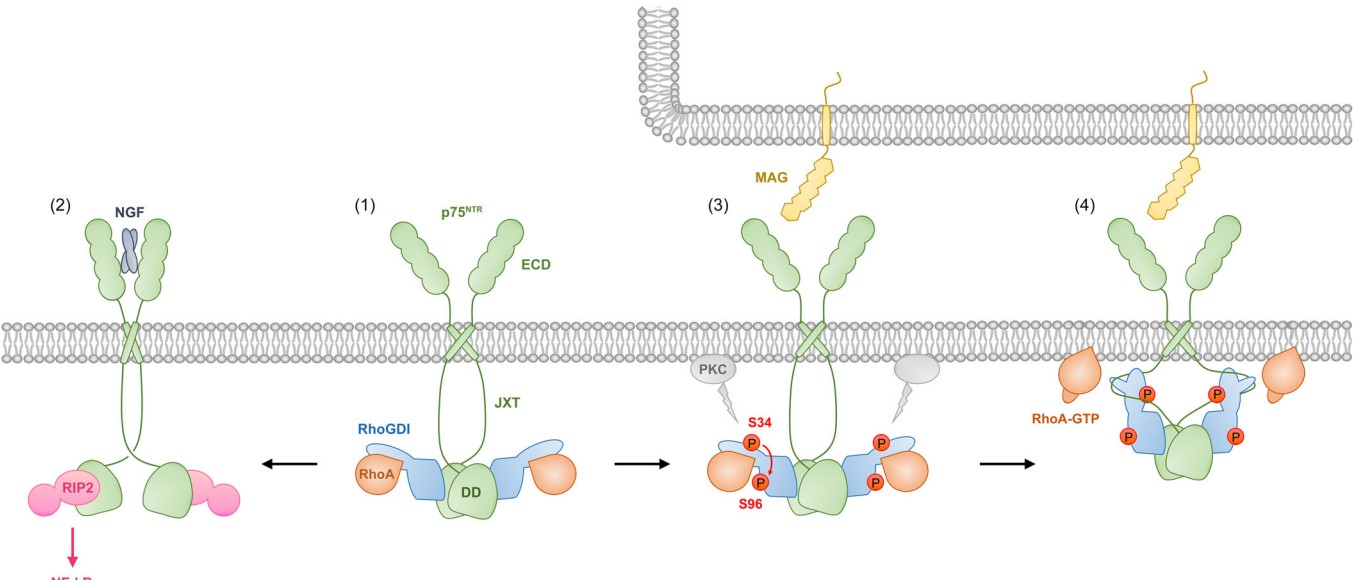

**Figure 8. Schematic of MAG-induced PKC-mediated RhoGDI phosphorylation and RhoA activation through p75[NTR].**

(1) Under basal conditions, RhoGDI CTD interacts with the p75[NTR] DD and recruits RhoA to the proximity of the plasma membrane. (2) NGF binding to p75[NTR] induces conformational changes in the receptor that weaken DD-DD interactions allowing recruitment of RIP2 and enhancement of NF-kB activity. (3) Acting from a nearby myelin membrane, MAG induces activation of plasma membrane-associated PKC which phosphorylates Ser[34] in RhoGDI NTD and subsequently Ser[96] in the CTD. For simplicity, MAG-binding subunits NgR and Lingo-1 were omitted from the diagram. (4) Phosphorylated Ser[34] interacts with Lys[303] of p75[NTR] JXT domain and RhoA translocates to the plasma membrane where it is loaded with GTP and activated. RhoGDI is shown here to interact in "*trans*" with p75[NTR] dimers, such that NTD and CTD of the same RhoGDI molecule bind to JXT and DD, respectively, of different p75[NTR] molecules in the receptor dimer, thereby locking the DD homodimer in a conformation that is no longer capable of recruiting RIP2.

plasma membrane, which considerably stabilizes and sustains its activity, even after dissipation of the initial stimuli (Huang, 1989). This mechanism is predominantly important in neurons, as they typically experience continued waves of Ca[2+] and DAG signaling. It is therefore also possible that p75[NTR] simply facilitates the initial recruitment of RhoGDI to the proximity of the plasma membrane where it can come into contact with active PKC molecules which in turn further stabilize RhoGDI interaction with the receptor through phosphorylation. In this regard, it is interesting to note that, even upon strong stimulation of PKC activity with PMA, RhoGDI was not phosphorylated and RhoA failed to be activated in knock-out CGNs lacking p75[NTR], supporting the role of the receptor in recruitment of the RhoGDI/RhoA complex to plasma membrane compartments. In the absence of strong PKC stimuli, NGF treatment of CGNs reduced RhoGDI Ser[96] phosphorylation below detection levels, suggesting active de-phosphorylation induced by NGF through p75[NTR]. On the other hand, NGF was unable to induce de-phosphorylation of Ser[96] in the presence of MAG or PMA. The identity of the phosphatases involved in RhoGDI de-phosphorylation is unclear. Fas-associated phosphatase-1 (FAP-1) has been shown to associate with the intercellular domain of p75[NTR] (Irie et al, 1999). However, FAP-1 has been described as a protein tyrosine phosphatase, and it is currently unknown whether it may also have phosphatase activity against phospho-Ser residues.

Phosphorylation of Ser[34] is critical for the effects of MAG on growth cone stability, neurite outgrowth, and apoptosis in CGNs, and establishes a novel mechanism for gating growth/collapse and survival/death outcomes regulated by p75[NTR] in primary

neurons. In addition to MAG, other myelin-derived proteins and peptides, including Nogo-66 and Oligodendrocyte Myelin Glyco-protein (OMgp), also utilize p75[NTR] as signaling receptor in complex with NgR and Lingo-1 (Wang et al, 2002a; Wong et al, 2002). All these proteins can induce growth cone collapse, inhibit neurite outgrowth and block nerve regeneration (Filbin, 2003; Wang et al, 2002b; GrandPré et al, 2000). Several studies have indicated that blockade or interference with the activities of myelin inhibitors enhances various forms of axon and nerve regeneration (He and Koprivica, 2004; Buchli et al, 2007). Our present findings suggest that, like MAG, PKC-mediated phosphor-ylation of RhoGDI Ser[34] may play a critical role in the activities of all myelin inhibitors that signal through p75[NTR]. It is thus conceivable that strategies aimed at disrupting this interaction, be it by interference with Ser[34] phosphorylation or its binding to Lys[303], may contribute to ameliorate the regeneration capacity of damaged axons.

In summary, our study presents unexpected new insights into the interaction of RhoGDI with p75[NTR] leading to RhoA activation, as well as detailed biochemical understanding of its regulation by PKC-mediated phosphorylation. This provides a mechanism to gate axon growth/collapse and cell survival/death decisions upon simultaneous encounter with neurotrophins and myelin-derived ligands. The notion of competition between downstream effectors for interaction with receptor intracellular domains, in turn dictated by the nature of incoming ligands, may be applicable to other receptors that engage different ligands, effectors and signaling pathways.

# Methods

## Plasmids, molecular cloning, and mutagenesis

A summary of all the plasmids used is presented in Table EV2. pcDNA5 FRT/TO mouse p75[NTR] expressing vector (DU31136) was purchased from MRC PPU reagents and services, Dundee University. Mouse flag-RhoGDI construct was obtained from Sinobiologicals. Mouse Myc-RIP2 and Myc-TRAF6 were obtained from Origene. All high-fidelity PCR was performed using NEB Q5 polymerase and all subcloning was done using NEBuilder HiFi DNA Assembly (NEB). Point mutations and deletions in mouse p75[NTR] and RhoGDI were introduced using NEB Q5 polymerase. PCR primers were synthesized by Integrated DNA Technologies (IDT) and dissolved in deionized distilled water. PCR was performed using 0.1 ng plasmids as DNA templates in a 25 μL PCR reaction mixture containing 12.5 μL Q5 polymerase master mix (2×) and 1.25 μL of 10 μM forward and reverse primers. PCR products were purified with a DNA purification kit (NEB) and were then phosphorylated using T4 polynucleotide kinase (NEB) according to the manufacturer's manual. DNA ligation was performed in the same reaction mixture containing the phosphorylated samples using 5U T4 DNA ligase (NEB). A transformation assay was performed using 3 μL DNA ligation product in DH5alpha bacterial cells and the positive clones were verified by Sanger sequencing.

## Mice

C57BL/6 mice were used for primary culture studies. Wild type and p75[NTR] knockout mice (Lee et al, 1994) of both sexes in C57BL/6 background were used for primary culture studies. All animals were housed in the National University of Singapore Comparative Medicine vivarium in a 12-h light/dark cycle and fed a standard chow diet. All animal procedures were approved by the National University of Singapore Institutional Animal Care and Use Committee (IACUC approval number 2020-01227).

## Cell line culture and transient transfection

NIH3T3 and HEK293 cells were cultured in Dulbecco's modified Eagle medium (DMEM) (Gibco) plus 10% fetal bovine serum (Life Technologies) supplemented with penicillin-streptomycin (Life Technologies). Cell lines were both negative for mycoplasma infection. The cells were transfected with the polyethylenimine (PEI) method. Briefly, cells were plated in a 10 cm tissue culture dish at a confluency of $2 \times 10^6$ cells/10 cm dish in normal growth media with antibiotics. Twenty-four hours after plating, transfection mix was prepared by mixing 10 μg of plasmid with 20 μg of PEI (1 mg/ml) in DMEM. The transfection mix was left to stand at room temperature followed by addition dropwise into culture plates. 24 h post-transfection, the cells were used for experiments.

## RhoGDI knock-down

Mouse endogenous RhoGDI knockdown was performed in NIH3T3 cells using Dicer Short Interfering RNAs purchased from IDT which target exon 4 of RhoGDI. The interfering RNA against RhoGDI was introduced in NIH3T3 cell lines by transfection with lipofectamine RNAiMAX. Successful knockdown of mouse endogenous RhoGDI was confirmed by western blot analysis. The sequences of the Dicer Short Interfering RNAs (mm.Ri.Arhg-dia.13.3) are as follows: SEQ1: rGrGrU rGrUrG rGrArG rUrArC rCrGrG rArUrA rArArA rArUC T; SEQ2: rArGrA rUrUrU rUrUrA rUrCrC rGrGrU rArCrU rCrCrA rCrArC rCrUrU.

## Primary culture of cerebellar granule neurons (CGNs)

P7 mouse cerebella were separated from the rest of the brain and placed into a plate with cold 1xHBSS buffer. Whole cerebella were digested with trypsin for extraction of CGNs. CGNs were plated at a density of 40,000 cells per coverslip coated with poly-L-lysine (PLL, Sigma) in a 24-well plate (ThermoScientific) in Neurobasal Medium (Gibco), supplemented with 10% Horse serum (Gibco), 25 mM KCl (Sigma), 1 mM glutamine (Gibco), and 2 mg/mL gentamicin (Invitrogen). Next day, the culture media was replaced with fresh Neurobasal media without horse serum supplemented with 25 mM KCl (Sigma), 1 mM glutamine (Gibco), and 2 mg/mL gentamicin (Invitrogen). To collect protein for immunoblotting, neurons were cultured at a higher density ($2 \times 10^6$ neurons per well) in a 6-well plate. The neurons were then treated for 30 min (with 25 μg/ml MAG or 50 ng/ml NGF) after 2 DIV. Mature NGF (N-100) was obtained from R&D Systems. Mouse MAG-Fc (51398-M02H) was obtained from SinoBiologicals. Treatment of CGNs with either 1 μM Gö6976 or 2 μM PMA was carried out overnight. In some experiments, expression plasmid constructs were introduced into CGNs by Neon electroporation according to manufacturer instructions (Life Technologies). Briefly, P7 CGNs were resuspended at high cell density ($4 \times 10^6$/100 μL) in electroporation Buffer-R. Subsequently, 5–7 μg of salt-free plasmid DNA was added and Neon tube with 3 mL Electrolytic Buffer E2 was set up. Electroporation was performed in a 100 μl tip with 2 pulses of 20 ms width at 1200 V. Immediately after electroporation and the CGNs were plated into a 6-well plate.

## Lentivirus production and transduction of CGNs

Lentiviruses expressing p75[NTR] wild type, K303A mutant, RhoGDI wild type, and RhoGDI S34D mutant were produced by transfecting HEK293T cells with pHAGE-IRES-eGFP constructs together with standard packaging vectors (pCMV-dR8.74 and pCMV-VSVG) by PEI based transfection method followed by ultra-centrifugation-based concentration of viral particles. Virus titer (T) was calculated based on the infection efficiency for HEK293T cells, where $T = (P*N)/(V)$, T = titer (TU/μL), P = % of infection positive cells according to the fluorescence marker, N = number of cells at the time of transduction, V = total volume of virus used. Note TU stands for transduction unit. One day before transduction, CGNs were plated on PLL treated 24 well cell culture plate. On the next day, the neurobasal medium was aspirated and CGNs were incubated with 200 μL of viral particles containing neurobasal media overnight. Next day the medium was removed and replaced with fresh neurobasal media and the cells were incubated for 1–2 days before measuring gene activity.

## Co-immunoprecipitation and Western blotting

Cells were washed 3 times with sterile ice-cold PBS then lysed in lysis buffer (50 mM Tris/HCl pH 7.5, 1 mM EDTA, 270 mM Sucrose, 1% (v/v) Triton X-100, B mercaptoethanol and 60 mM Octyl β-glucoside) containing protease and phosphataseinhibitors.

Lysates were cleared at $10,000 \times g$ pellet for 1 min at 4 °C. When indicated, control or phospho-Ser34 RhoGDI peptide (Cys-PAQKpSIQEI) was added to the lysate at 30 µM and incubated with lysate for 30 min at 4 °C prior to addition of antibody. For immunoprecipitation, 1 µg of p75NTR antibody was added to a total of 500 µg protein (1 mg/mL in lysis buffer) and was incubated overnight on an orbital shaker at 4 °C. On the next day, 25 µL bed-volume of ethanolamine-treated protein G sepharose resin (Pierce) was added to the lysate, and mixtures were incubated for 2 h at 4 °C with rotation. The resin was washed once with wash buffer 1 (50 mM Tris, 250 mM Sucrose, 5 mM MgCl$_2$, 0.15 M NaCl, 2% Igepal, 200 mM ethanolamine, pH 10.5) and washed 3× with the wash buffer 2 (20 mM Tris, pH 7.5, 250 mM Sucrose, 5 mM MgCl$_2$, 0.15 M NaCl, 2% Igepal), then boiled in 30 µL 2× SDS sample buffer, and eluted proteins were analyzed by SDS/PAGE followed by Western blotting. 20 µg of clarified total cellular lysate was loaded per well to be separated on an SDS-PAGE followed by western blotting on polyvinylidene difluoride membranes membrane. Western blots were probed using HRP conjugated secondary antibody, and detected using ECL detection kit (Cell Signaling). A summary of antibodies used is presented in Table EV3.

## RhoA and NF-kB activation assays

On the day of harvest, cells were rinsed immediately in ice-cold phosphate-buffered saline (PBS, Sigma) prior to lysate collection. Cells were stored in G-LISA lysis buffer with manufacturer's protease inhibitors (Cytoskeleton). The absorbance at 490 nm measures the active GTP-bound RhoA in the neuronal lysates that have bound to the Rhotekin RhoA binding domain immobilized to the 96-well plate. Protein concentrations were determined using Precision Red Advanced Protein Assay Reagent with absorbance at 600 nm measured spectrophotometer. Lysates were incubated in wells for 30 min at 4 °C with shaking, then rinsed and sequentially incubated with primary antibodies to RhoA followed by HRP secondary antibodies (Cytoskeleton), and the colorimetric reaction was performed for 15 min at 37 °C, which was measured using a microplate plate reader (BioTek). To assess NF-kB activation, we used an assay (Active Motif products, #43296) in which a colorimetric readout measures activated NF-kB p65 subunit on whole cell extracts. Whole cell lysates from cultures of CGNs were applied to a 96-well plate to which oligonucleotide containing an oligonucleotide containing a p65 NFκB consensus DNA binding site has been immobilized. The colorimetric readout at 450 nm measures activated NF-kB p65 on whole cell extracts bound to the oligonucleotide immobilized to the 96-well plate.

## Protein purification

The cDNAs of human p75NTR JXT (K273-D332, ~6.6 kDa) and RhoGDI NTD$^{S34D}$ (A2-A60, ~6.8 kDa) were cloned in a pET-32a-derived vector, respectively. The His-tagged target proteins were expressed in E. coli BL21 (DE3) in Luria broth (LB) medium and purified by nickel-nitrilotriacetic acid (Ni-NTA) affinity chromatography. His tag was removed by tobacco etch virus (TEV) protease cleavage. The proteins were further purified by gel filtration (HiLoad 16/600 Superdex 75 pg, Cytiva) and ion exchange chromatography (p75NTR-JXT: MonoS 10/100 GL, Cytiva; RhoGDI NTD$^{S34D}$: MonoQ 10/100 GL, Cytiva). The purity of the target proteins was verified by SDS-PAGE. The molecular weights (MWs)

of the target proteins were verified by MALDI-TOF mass spectra. $^{15}$N-labeled sample was expressed in M9 minimum medium using $^{15}$N-NH4Cl as the sole nitrogen source. The purification method was the same as that of the proteins expressed in LB medium. The purified p75NTR JXT domain and RhoGDI NTD$^{S34D}$ were buffer-exchanged to 10 mM PIPES pH 6.8, respectively. NMR titration of unlabeled protein to the $^{15}$N-labeled sample was performed at 28 °C. $^{15}$N, $^{13}$C-labeled samples were expressed in M9 medium using $^{15}$N-NH4Cl and $^{13}$C-glucose as the sole carbon and nitrogen source, respectively. $^{15}$N,$^{13}$C-labeled p75NTR JXT, and unlabeled RhoGDI NTD$^{S34D}$ were mixed together at a mole ratio of 1:3. Similarly, $^{15}$N, $^{13}$C-labeled RhoGDI NTD$^{S34D}$, and unlabeled p75NTR JXT were also mixed at the mole ratio of 1:3. The concentration of $^{15}$N,$^{13}$C-labeled proteins in the final NMR sample were ~0.8 mM.

## NMR spectroscopy and structure calculations

NMR data were acquired on a Bruker 800 MHz NMR spectrometer with a cryogenic probe. In order to minimize protein degradation, NMR spectra for structural determination were acquired at 20 °C. The experiments include 2D $^1$H, $^{15}$N-heteronuclear single quantum coherence spectroscopy (HSQC), 2D $^1$H, $^{13}$C-HSQC, 3D HNCACB, 3D CACB(CO)NH, 3D CC(CO)NH, 3D H(CCCO)NH and multiple quantum $^{13}$C total correlation spectroscopy (MQ-CCH-TOCSY). NMRPipe and NMRView were used to process and analyze the spectra, respectively (Delaglio 1995, Johnson 2004). Intramolecular nuclear Overhauser effects (NOEs) were obtained from 3D $^{15}$N, $^{13}$C- and $^{13}$C, $^{13}$C-NOESY. Intermolecular NOEs were assigned based on 3D $^{13}$C, $^{15}$N-filtered NOESY. CYANA was used to calculate the complex structure in CYANA Ten conformers with the lowest CYANA target function values were energy-minimized in the AMBER force field (Case et al, 2005). PROCHECK-NMR was used to evaluate the quality of the structures (Laskowski et al, 1996). The coordinates of RhoGDI-NTD$^{S34D}$:p75NTR-JXT complex were deposited in the Protein Data Bank with PDB ID: 8X8T.

## In situ proximity ligation assay (PLA)

The in situ proximity ligation assay (Duolink® In Situ Detection Reagents Orange, Duolink® In Situ PLA® Probe Anti-Mouse MINUS, Duolink® In Situ PLA® Probe Anti-Rabbit PLUS, and Duolink® In Situ Wash Buffers, Fluorescence reagents, all from Sigma Aldrich) was performed in the following manner. Cells on coverslips were fixed with 4% PFA for 20 min at RT and washed three times with PBS. Cells were permeabilised in 0.1% Triton X-100 for 30 min at RT and washed with PBS-T (0.05% Tween 20) three times, for 5 min each time. Subsequently, one drop of Duolink II Blocking Solution was added directly to the coverslips and were incubated for 1 h at RT. Afterward, antibodies against RhoGDI (mouse antibody) and against p75NTR (ANT-011, rabbit antibody) were diluted in antibody diluent and added as 40 µL to each coverslip. The slides were incubated overnight at 4 °C. The slides were washed twice with PBS-T with gentle agitation. Next day, 40 µL of PLA probes mix (10 µL PLA probe PLUS stock + 10 µL PLA probe MINUS stock + 30 µL antibody diluent) was added to each slide and the samples were incubated for 1 h at 37 °C in a humidified chamber. The slides were then washed twice with wash buffer A for 5 min with gentle agitation. The ligase was diluted 1 in 40 ratio in 1X Duolink II Ligation solution. 40 µL of Duolink II

Ligation solution containing ligase were added to each slide and incubated for 30 min at 37 °C. The slides were again washed twice with wash buffer A for 2 min with gentle agitation. Meanwhile, the Duolink II Amplification stock (5X) was diluted 1 in 5 in high purity water to obtain a 1X Duolink II Amplification solution. The polymerase was then diluted 1 in 80 in the Duolink II Amplification solution and 40 µL added to each slide. Incubation was performed for 2 h at 37 °C in a humidified chamber. The slides were washed twice with wash buffer B for 10 min with gentle agitation followed by another wash with 0.01X wash buffer B diluted in high purity water for 2 min with gentle agitation. The slides were then washed once with wash buffer A for 1 min and the slides were mounted using Duolink® In Situ mounting medium with DAPI (Sigma). Images were acquired with Leica confocal microscope.

### Growth cone collapse, neurite outgrowth, and cleaved caspase 3 immunofluorescence

CGNs were transduced with the indicated lentivirus on day 2 in vitro and then cells were fixed for immunofluorescence by 4% paraformaldehyde on day 5 in vitro. The fixed and permeabilized cells were then labeled with antibodies against beta-Tubulin (TUJ-1). Growth cones in cultured CGNs were observed at ×60 magnification. The total number of growth cones in all cells after each treatment was counted. Only growth cones with both filopodia and lamellipodia were counted. The percentage of collapsed growth cones was calculated as the percentage of decrease of the number of growth cones after each treatment. Neurite outgrowth was evaluated by imaging GFP and assessing the length of the longest neurite in each neuron. For assessing cleaved caspase 3, neurons were transduced with the indicated lentivirus on day 2 in vitro and then cells were fixed for immunofluorescence by 4% paraformaldehyde on day 5 in vitro. The fixed and permeabilized cells were then labeled with antibodies against cleaved caspase 3 (Cell Signaling Technology; 9761; 1:500 dilution) and counterstained with DAPI. Images were taken using a Leica fluorescence microscope and quantified to obtain percentage cells positive for cleaved caspase 3 relative to DAPI.

### Image analysis

Image processing and quantification were performed with the software ImageJ (NIH; Imaging Processing and Analysis in Java; https://imagej.nih.gov/ij). The NeuronJ plugin for FIJI was used for neurite length analysis. Averaged neurite lengths of each group were normalized to that of wild-type untreated CGNs. For image analysis of PLA, the soma was manually outlined for each cell and number of puncta per soma was counted with ImageJ. The number of puncta in the processes was calculated by counting total number of puncta per image. The number of puncta was normalized by number of cells to obtain mean PLA puncta/cell. The threshold was set automatically using ImageJ for each image and kept constant as the puncta in the soma were measured separately. Microscope settings were kept constant for all images to enable direct comparison. Quantification was performed on one set of experiments, when all staining were performed at the same time.

### Statistical analysis

Data were analyzed using GraphPad Prism 8 software. Two-way ANOVA followed by Tukey's multiple comparison test was used to compare the amount of RhoA activation using GLISA assay. One-way ANOVA followed by Bonferroni post hoc test was performed to compare the PLA puncta in soma/processes of lentivirus transducted primary neurons (CGNs).

## Data availability

The NMR coordinates of the RhoGDI NTD/p75NTR JXT complex have been deposited in Protein Data Bank with accession number 8X8T.

## Peer review information

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

## Acknowledgements

We thank Yajing Duan (Tianjin University) for preliminary protein studies, Ket Yin Goh and Jing-song Fan (NUS) for management and technical assistance,

Jingfeng Zhang, (IAPMST, China) and Shuo Shi and Shuo Shi (Nankai University) for help with NMR titration. Support for this research was provided by grants to CFI from Swedish Cancer Society (Cancerfonden, contract nr. 222135Pj01H), Swedish Research Council (Vetenskapsrådet, contract nr. 2020-01923_3), National Research Foundation of Singapore (NRF, contract nr. NRF2019-NRF-ISF003-3141) and to ZL from National Natural Science Foundation of China (contract nr. 21974093).

## Author contributions

**Ajeena Ramanujan**: Formal analysis; Investigation; Methodology; Dr. Ajeena Ramanujan performed and analyzed all experiments in the manuscript except the NMR structure. She should therefore be considered the main first author of this paper. **Zhen Li**: Investigation; Zhen Li solved the NMR structure reported in the paper following instructions from his supervisor Dr. Zhi Lin. Zhen Li had no participation in experimental design or conceptualization aspects of the paper. He should therefore be considered second to Dr. Ramanujan in ranking of contributions. **Yanchen Ma**: Investigation. **Zhi Lin**: Formal analysis; Supervision; Funding acquisition. **Carlos F Ibáñez**: Conceptualization; Formal analysis; Supervision; Funding acquisition; Writing—original draft; Project administration; Writing—review and editing.

## Funding

## Disclosure and competing interests statement

The authors declare no competing interests.

# Expanded View Figures

**Figure EV1.  Solution structure of the complex between the p75^NTR JXT and RhoGDI NTD^S34D.**

(**A**) SDS PAGE profiles of purified RhoGDI NTDS34D and p75NTR JXT domain. (**B**) [1H-15N] HSQC spectra of 15N-RhoGDI NTD in the absence (black) and presence (red) of p75^NTR JXT at 28 °C. The concentration of RhoGDI NTD and p75^NTR JXT was 0.3 mM and 1.5 mM, respectively. The backbone and side chain cross peaks undergoing significant chemical shift changes are labeled. * indicates unassigned cross-peak. (**C**) Representative slices from the 13C, 13C-filtered 3D NOESY spectrum. (**D**) Interaction diagram of complex interface created by LigPlot+. Amino acid residues involved in electrostatic interactions are shown as a ball-and-stick model, and residues involved in hydrophobic interactions are shown as an "eyebrow" shape. The black dashed line indicates the interaction interface and the green dashed line connects the residues involved in the electrostatic interactions. Molecule A represents RhoGDI NTDS34D, and B represents p75NTR JXT. (**E**) Schematic of *cis* and *trans* modes of RhoGDI interaction with the intracellular domains of the p75NTR dimer. (**F**) "Top view" of structural model of quaternary complex between p75^NTR death domain (DD, brown) and RhoGDI (cyan). The model is based on the solution structure of the RhoGDI CTD bound to p75^NTR DD (Lin 2015) and the structure of the full-length RhoGDI (from PDB ID:4F38). Arrows denote the spatial proximity of the RhoGDI N-terminal domain (NTD) to the N-terminus (N) of the p75^NTR DD that engages the C-terminal domain (CTD) of the second RhoGDI protomer. (**G**) Crystal structure of RhoGDI:RhoA complex (PDB ID: 4F38). Ser34 from the RhoGDI NTD is in close proximity to Glu121 from the RhoGDI CTD. The distances between side chain oxygen atoms of these two residues are ~5 Å.

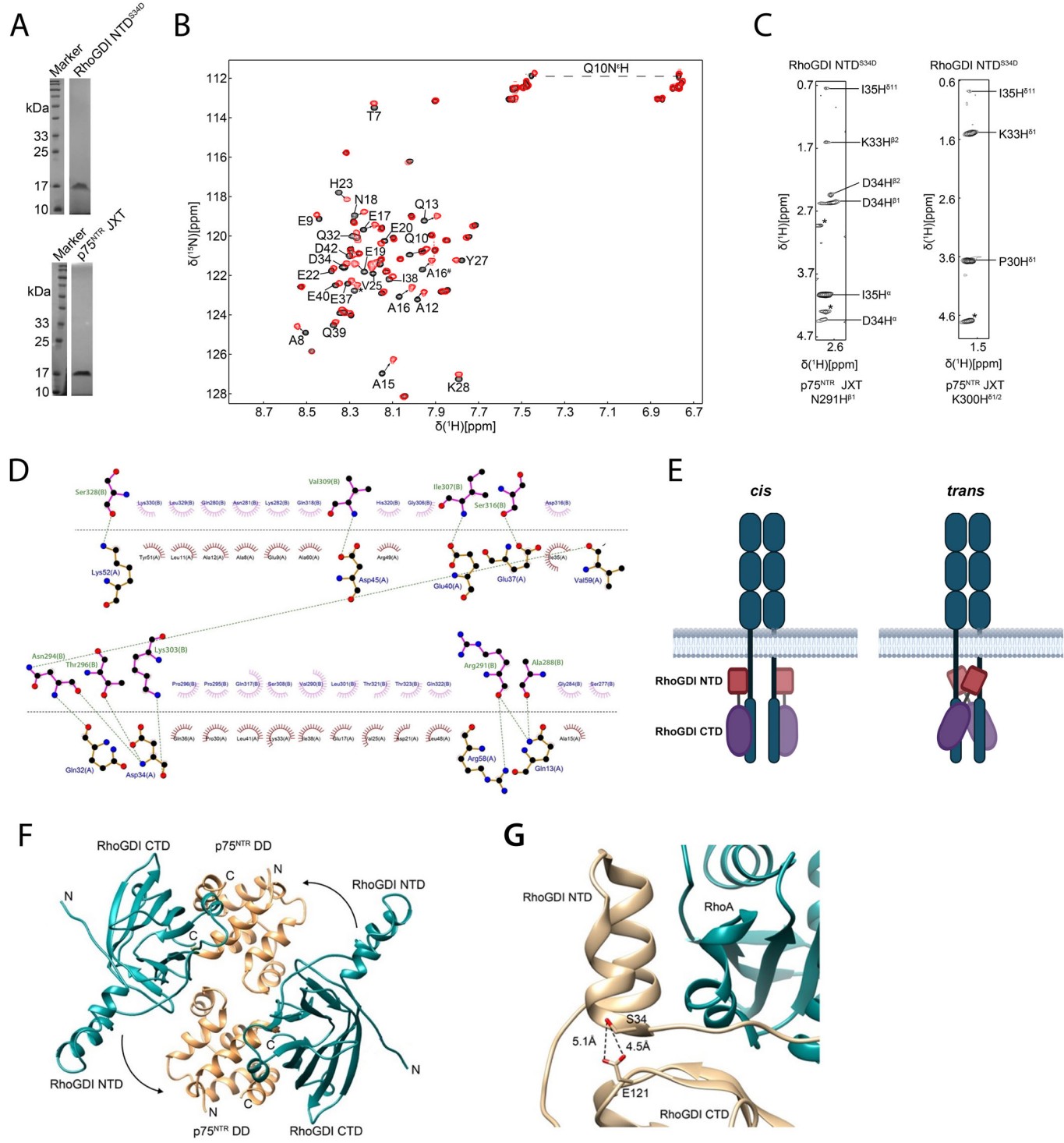

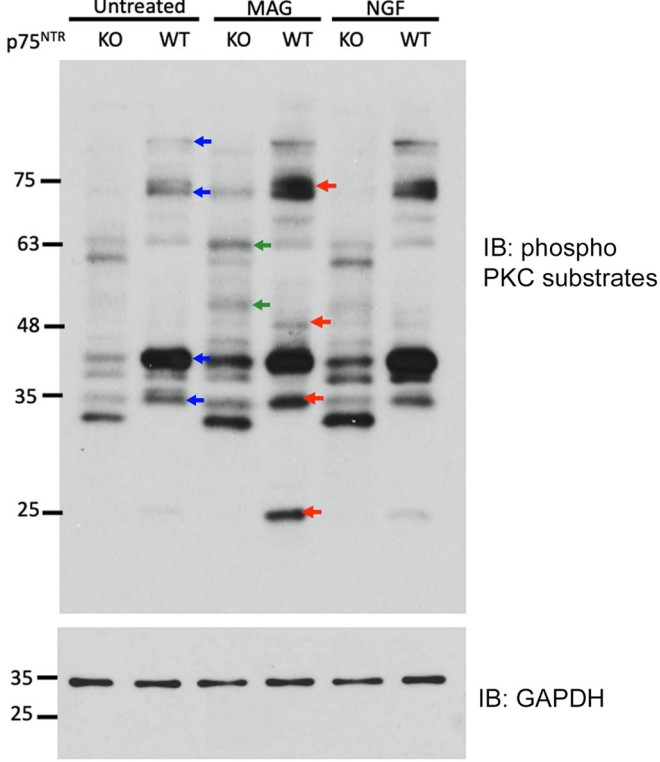

**Figure EV2. Western blot of phospho PKC substrates in total CGN extracts.**

Western blot of phospho PKC substrates in total cell extracts of P7 p75NTR knock out (KO) or p75NTR wild type (WT) CGNs following treatment with MAG (25ug/ml) or NGF (100 ng/ml) for 30 min. Blue arrows denote protein species showing de-novo or enhanced phosphorylation in CGNs expressing p75NTR compared to KO cells. Green arrows denote species showing increased phosphorylation after MAG treatment independently of p75NTR. Red arrows denote species showing increased phosphorylation after MAG treatment in a p75NTR-dependent manner.

