## [Peer Review File · EMBO Reports]

RhoGDI phosphorylation by PKC promotes its interaction with death receptor p75NTR to gate axon...

Ajeena Ramanujan, Zhen Li, Yanchen Ma, Zhi Lin, and Carlos Ibanez
DOI: 10.15252/embr.202358305

Corresponding author(s): Carlos Ibanez (carlos.ibanez@pku.edu.cn)

Review Timeline:

Submission Date:	11th Oct 23
Editorial Decision:	10th Nov 23
Revision Received:	6th Dec 23
Editorial Decision:	19th Dec 23
Revision Received:	21st Dec 23
Accepted:	5th Jan 24

Transaction Report:

Dear Prof. Ibanez

Thank you for the submission of your research manuscript to our journal. We have now received the full set of referee reports that is copied below.

As you will see, the referees consider the findings interesting and overall well supported by the data but they also have a number of suggestions how to further improve your study that need to be addressed.

Given these constructive comments, we would like to invite you to revise your manuscript with the understanding that the referee concerns (as detailed above and in their reports) must be fully addressed and their suggestions taken on board. Please address all referee concerns in a complete point-by-point response. Acceptance of the manuscript will depend on a positive outcome of a second round of review. It is EMBO Reports policy to allow a single round of revision only and acceptance or rejection of the manuscript will therefore depend on the completeness of your responses included in the next, final version of the manuscript.

We realize that it is difficult to revise to a specific deadline. In the interest of protecting the conceptual advance provided by the work, we recommend a revision within 3 months (February 10, 2024). Please discuss the revision progress ahead of this time with the editor if you require more time to complete the revisions.

I am also happy to discuss the revision further via e-mail or a video call, if you wish.

*****IMPORTANT NOTE:

We perform an initial quality control of all revised manuscripts before re-review. Your manuscript will FAIL this control and the handling will be delayed IN CASE the following APPLIES:

- 1) A data availability section providing access to data deposited in public databases is missing. If you have not deposited any data, please add a sentence to the data availability section that explains that.
- 2) Your manuscript contains statistics and error bars based on $n=2$. Please use scatter blots in these cases. No statistics should be calculated if $n=2$.

When submitting your revised manuscript, please carefully review the instructions that follow below. Failure to include requested items will delay the evaluation of your revision.*****

- 1) a .docx formatted version of the manuscript text (including legends for main figures, EV figures and tables). Please make sure that the changes are highlighted to be clearly visible.
- 2) individual production quality figure files as .eps, .tif, .jpg (one file per figure). Please provide all figures in portrait orientation. You can download our Figure Preparation Guidelines (figure preparation pdf) from our Author Guidelines pages <https://www.embopress.org/page/journal/14693178/authorguide> for more info on how to prepare your figures.
- 3) a .docx formatted letter INCLUDING the reviewers' reports and your detailed point-by-point responses to their comments. As part of the EMBO Press transparent editorial process, the point-by-point response is part of the Review Process File (RPF), which will be published alongside your paper.
- 4) a complete author checklist, which you can download from our author guidelines (<<https://www.embopress.org/page/journal/14693178/authorguide>>). Please insert information in the checklist that is also reflected in the manuscript. The completed author checklist will also be part of the RPF.
- 5) Please note that all corresponding authors are required to supply an ORCID ID for their name upon submission of a revised manuscript (<<https://orcid.org/>>). Please find instructions on how to link your ORCID ID to your account in our manuscript tracking system in our Author guidelines (<<https://www.embopress.org/page/journal/14693178/authorguide#authorshipguidelines>>)
- 6) We replaced Supplementary Information with Expanded View (EV) Figures and Tables that are collapsible/expandable online. A maximum of 5 EV Figures can be typeset. EV Figures should be cited as 'Figure EV1, Figure EV2' etc... in the text and their respective legends should be included in the main text after the legends of regular figures.

- For the figures that you do NOT wish to display as Expanded View figures, they should be bundled together with their legends in a single PDF file called *Appendix*, which should start with a short Table of Content including page numbers. Appendix figures should be referred to in the main text as: "Appendix Figure S1, Appendix Figure S2" etc. See detailed instructions regarding expanded view here:

<<https://www.embopress.org/page/journal/14693178/authorguide#expandedview>>

7) Please note that a Data Availability section at the end of Materials and Methods is now mandatory. In case you have no data that requires deposition in a public database, please state so instead of refereeing to the database.

See also < <https://www.embopress.org/page/journal/14693178/authorguide#dataavailability>>. Please note that the Data Availability Section is restricted to new primary data that are part of this study.

Additional information on source data and instruction on how to label the files are available

<<https://www.embopress.org/page/journal/14693178/authorguide#sourcedata>>.

10) Figure legends and data quantification:

- the name of the statistical test used to generate error bars and P values,
- the number (n) of independent experiments (please specify technical or biological replicates) underlying each data point,
- the nature of the bars and error bars (s.d., s.e.m.)

- If the data are obtained from n {less than or equal to} 5, show the individual data points in addition to the SD or SEM.

- If the data are obtained from n {less than or equal to} 2, use scatter blots showing the individual data points.

11) Our journal encourages inclusion of *data citations in the reference list* to directly cite datasets that were re-used and obtained from public databases. Data citations in the article text are distinct from normal bibliographical citations and should directly link to the database records from which the data can be accessed. In the main text, data citations are formatted as follows: "Data ref: Smith et al, 2001" or "Data ref: NCBI Sequence Read Archive PRJNA342805, 2017". In the Reference list, data citations must be labeled with "[DATASET]". A data reference must provide the database name, accession number/identifiers and a resolvable link to the landing page from which the data can be accessed at the end of the reference. Further instructions are available at <<https://www.embopress.org/page/journal/14693178/authorguide#referencesformat>>.

12) All Materials and Methods need to be described in the main text. We would encourage you to use 'Structured Methods', our new Materials and Methods format. According to this format, the Materials and Methods section should include a Reagents and Tools Table (listing key reagents, experimental models, software and relevant equipment and including their sources and relevant identifiers) followed by a Methods and Protocols section in which we encourage the authors to describe their methods using a step-by-step protocol format with bullet points, to facilitate the adoption of the methodologies across labs.

More information on how to adhere to this format as well as downloadable templates (.doc or .xls) for the Reagents and Tools Table can be found in our author guidelines: <

<https://www.embopress.org/page/journal/14693178/authorguide#manuscriptpreparation>>.

<<https://www.embopress.org/doi/10.15252/msb.20178071>>.

13) As part of the EMBO publication's Transparent Editorial Process, EMBO Reports publishes online a Review Process File to

accompany accepted manuscripts. This File will be published in conjunction with your paper and will include the referee reports, your point-by-point response and all pertinent correspondence relating to the manuscript.

Yours sincerely,

Referee #1:

The manuscript by Ramanujan et al. presents a compelling study on the regulation of p75NTR signaling and its implications for neuron survival and axon growth. The paper is well-constructed, with clear demonstrations of how p75NTR's interactions with various ligands, such as NGF and MAG, and downstream effectors like RhoGDI and RIP2, lead to different neuronal outcomes. The use of NMR to elucidate the structure of the RhoGDI-p75NTR complex is particularly noteworthy and adds a valuable dimension to the understanding of receptor signaling. The novelty of the work is evident in its detailed elucidation of the mechanisms by which different ligands modulate p75NTR's engagement with downstream effectors. This contributes significantly to the field of neuroscience by providing insights into how neurons integrate multiple signals for making critical growth and survival decisions. Overall Assessment:

This study represents one of the most elegant explanations of p75NTR's dual functions and its ability to switch between growth and survival signaling pathways. The research is methodologically sound, and the findings are clearly presented and well-supported by the data. With the minor revisions suggested below, I am in favor of accepting this manuscript for publication. The study will undoubtedly be a valuable addition to the field.

Minor Points:

- * While the paper is strong in its current form, a more explicit discussion on the implications of the findings for potential therapeutic targets would enhance its impact. This could involve a deeper analysis of how the detailed mechanism might be exploited to modulate neuron survival and axon growth in pathological conditions.
- * In Figure 7b, the appearance of k303A axons suggests more degeneration than other conditions. It would be helpful to clarify if this is a consistent observation or an artifact and to provide quantification for axonal integrity if available. Perhaps it simply requires a different representative image.
- * Expanding the summary in Figure 8 to show the relationship between Serine 34 and Serine 96 phosphorylation events would help in illustrating the interplay between these modifications more clearly.
- * A brief explanation of the NF- κ B and Rho GTP assays in the Results section would be useful for readers who may not be familiar with these techniques.
- * The first mention of NIH3T3 cells in the discussion of Figure 1 should include a note that these cells do not express endogenous p75NTR, as this is a crucial piece of information for the interpretation of the data. This information is currently introduced in the discussion of Figure 2 and would be more appropriately placed earlier in the text.

Referee #2:

This study focuses on the receptor p75NTR, a receptor that mediates signaling by several structurally unrelated ligands, and couples to signaling by several alternative pathways, via sometimes competitive binding of a variety of signaling adapter proteins. The study reveals specific mechanisms that dictate which signaling outcome prevails when the receptor encounters two ligands, MAG and NGF simultaneously. The investigators find that NGF promotes recruitment of RIP2 to p75NTR, displacing RhoGDI and thereby decreasing RhoA activity while increasing NF- κ B activation, while MAG induces PKC-

mediated phosphorylation of Ser34 of RhoGDI, promoting association of RhoGDI with p75NTR and increasing RhoA activation. The study provides structural data yielding insight into the mechanisms of these interactions. Further, the study reveals how association of these signaling proteins with the juxtamembrane domain of p75NTR imparts a more rigid structure on this otherwise unstructured domain, and providing a possible mechanism that allows coupling of structural changes in the receptors extracellular domain to structural changes in the intracellular domain

Overall, the paper is technically impressive and rigorous, well-written and interesting.

Major issues: I found only one serious short-coming in the presentation. The authors correctly note that the mechanism by which MAG promotes the association with p75NTR is unclear. However, what they fail to indicate is that part of this lack of clarity reflects the fact that MAG does not bind to p75NTR. Two MAG-binding proteins, NogoR and LRP1 have been reported to associate with p75NTR in a MAG-dependent way, thereby acting as coreceptors. The authors refer to MAG as a p75NTR ligand, but this is misleading. MAG is a ligand that activates p75NTR, but it is not a ligand that binds p75NTR. I know the authors are aware of this fact, and I suspect that they elected to avoid discussion of this complexity because none of their studies bear directly on whether either of these co-receptors is involved. However, I think failing to discuss this will ultimately be confusing to the readers.

I found a number of small textual errors that need to be corrected for clarity, as follows:

Page 6. "while Ser96 is dispensable for RhoGDI binding" should read "while Ser96 phosphorylation is dispensable for RhoGDI binding"

PAGE 6: "we generated phosphor-specific antibodies against phospho-Ser96" should read "we generated phospho-specific antibodies against phospho-Ser96". An auto-correct issue.

Page 8: "Mutant S34A RhoGDI-59 did not co-immunoprecipitated" should read "Mutant S34A RhoGDI-59 did not co-immunoprecipitate"

Page 13: "effects axon growth and neuron survival" should read "effects on axon growth and neuron survival"

Page 14: "Even within the same cell type, can p75NTR induce" should read "Even within the same cell type, p75NTR can induce"

PAGE 15: "mechanism by each PKC stimulates" should read "mechanism by which PKC stimulates"

Answers to reviewers' comments – EMBOR-2023-58305V1**Data added to the revised version**

We have added one piece of data to the revised version of our manuscript. The reviewers may recall that in the diagram presented in Figure 8 at the end of the paper, we mentioned the possibility that RhoGDI may interact in “trans” with p75^{NTR} dimers, such that NTD and CTD of the same RhoGDI molecule bind to JXT and DD, respectively, of different p75^{NTR} molecules in the receptor dimer. We have now been able to demonstrate this in a co-immunoprecipitation experiment using complementation between two different p75^{NTR} molecules carrying mutations in either JXT or DD, respectively. These data are now shown in Figure 6A along with explanatory graphics presented in Figures EV1E and F. The corresponding paragraph in the Results section also explains this in more detail.

We have also updated the solution structure of the RhoGDI NTD/p75^{NTR} JXT complex (Figure 5) after refinement of the model (improved parameters in Table EV1).

Reviewer #1

While the paper is strong in its current form, a more explicit discussion on the implications of the findings for potential therapeutic targets would enhance its impact. This could involve a deeper analysis of how the detailed mechanism might be exploited to modulate neuron survival and axon growth in pathological conditions.

A discussion of this point has been included (p. 18-19)

In Figure 7b, the appearance of k303A axons suggests more degeneration than other conditions. It would be helpful to clarify if this is a consistent observation or an artifact and to provide quantification for axonal integrity if available. Perhaps it simply requires a different representative image.

Axonal integrity was not compromised in this condition as verified by counterstaining for Tuj1). This is now indicated in the figure legend.

Expanding the summary in Figure 8 to show the relationship between Serine 34 and Serine 96 phosphorylation events would help in illustrating the interplay between these modifications more clearly.

The figure could not be expanded but was modified so as to denote the relationship between the two Ser residues.

A brief explanation of the NF-κB and Rho GTP assays in the Results section would be useful for readers who may not be familiar with these techniques.

This was now included in the Results and Methods sections.

The first mention of NIH3T3 cells in the discussion of Figure 1 should include a note that these cells do not express endogenous p75^{NTR}, as this is a crucial piece of information for the interpretation of the data. This information is currently introduced in the discussion of Figure 2 and would be more appropriately placed earlier in the text.

This was now included in the Results section.

Reviewer #2

I found only one serious short-coming in the presentation. The authors correctly note that the mechanism by which MAG promotes the association with p75^{NTR} is unclear. However, what they fail to indicate is that part of this lack of clarity reflects the fact that MAG does not bind to p75^{NTR}. Two MAG-binding proteins, NogoR and LRP1 have been reported to associate with p75^{NTR} in a MAG-dependent way, thereby acting as coreceptors. The authors refer to MAG as a p75^{NTR} ligand, but this is misleading. MAG is a ligand that activates p75^{NTR}, but it is not a ligand that binds p75^{NTR}. I know the authors are aware of this fact, and I suspect that they elected to avoid discussion of this complexity because none of their studies bear directly on whether either of these co-receptors is involved. However, I think failing to discuss this will ultimately be confusing to the readers.

We have modified the Introduction section to clarify this point. But we would like to note that, as far as we are aware, no evidence has been presented that MAG does not interact with p75^{NTR} in the complex with co-receptors NgR and Lingo-1, only that it does not in their absence (Yamashita et al, 2002). (Incidentally, this paper does not actually show this either, but refers to “unpublished preliminary experiments”. We will welcome any additional information the reviewer may have that would help to clarify this point.) In the GDNF/GFR α 1/RET and TGF β -TBR11-TBR1 complexes, the ligands (GDNF and TGF β) are unable to bind their signaling receptors (RET and TBR1/ALK5, respectively) in the absence of the binding co-receptors (GFR α 1 and TBR11, respectively), but cross-linking and x-ray crystallography studies have conclusively established that both ligands do make direct contacts with the signaling receptors. However, this binding is too weak to be detected with the isolated signaling receptors (Ibáñez, 2013; Derynck & Budi, 2019). Likewise, MAG may interact directly with p75^{NTR} through contacts that are too weak to be detected in the absence of the NgR and Lingo-1 co-receptors. In the absence of similar evidence, it cannot be concluded at present that MAG does not make direct contacts with p75^{NTR}.

We have also made all the corrections indicated below (and thank this reviewer for his/her thorough reading of our manuscript):

Page 6. "while Ser96 is dispensable for RhoGDI binding" should read "while Ser96 phosphorylation is dispensable for RhoGDI binding"

PAGE 6: "we generated phosphor-specific antibodies against phosphor-Ser96" should read "we generated phospho-specific antibodies against phospho-Ser96". An auto-correct issue.

Page 8: "Mutant S34A RhoGDI-59 did not co-immunoprecipitated" should read "Mutant S34A RhoGDI-59 did not co-immunoprecipitate"

Page 13: "effects axon growth and neuron survival" should read "effects on axon growth and neuron survival"

Page 14: "Even within the same cell type, can p75^{NTR} induce" should read "Even within the same cell type, p75^{NTR} can induce"

PAGE 15: "mechanism by each PKC stimulates" should read "mechanism by which PKC stimulates"

Manuscript number: EMBOR-2023-58305V2

Title: RhoGDI phosphorylation by PKC regulates its interaction with death receptor p75^{NTR} to gate axon...

Author(s): Ajeena Ramanujan, Zhen Li, Yan Chen Ma, Zhi Lin, and Carlos Ibanez

Dear Prof. Ibanez

Thank you for your patience while we have editorially reviewed your revised manuscript. I am now writing with an 'accept in principle' decision, which means that I will be happy to accept your manuscript for publication once a few minor issues/corrections have been addressed, as follows.

- Page 13, top: the reference to Figure 6A is incorrect when you describe the effect of NGF treatment on p75^{NTR}-RhoGDI interaction. I believe this should refer to Figure 6B. From there on all references to Figure 6 panels need to be checked. I think the addition of panel A has messed up the callouts.
- Please reduce the number of keywords to 5.
- Please update the 'Conflict of interest' paragraph to our new 'Disclosure and competing interests statement'. For more information see <https://www.embopress.org/page/journal/14693178/authorguide#conflictsofinterest>
- Please remove the Author Contributions from the manuscript file and make sure that the author contributions in our online submission system are correct and up-to-date. The information you specified in the system will be automatically retrieved and typeset into the article. You can enter additional information in the free text box provided, if you wish.
- Please note that our editorial policies do not allow to base conclusions on 'data not shown', which has been used on pages 9, 23, and in the legend of Figure 7b (Tuj1 counterstaining). Please either remove the conclusions or show the data, e.g. in the Appendix.
- Callouts are missing for the following figure panels: Fig 2E, Fig 2I, Fig 6P; Supplementary Information Table EV2 should be renamed to Table EV2; Figure S1A should be renamed to Appendix Figure S1A; S4C needs to be renamed to Appendix Figure S4C
- Appendix: the nomenclature needs correcting in the table of content and throughout the file: e.g. (Supplementary) Figure S1 should be Appendix Figure S1. All callouts in the text need to be checked/updated.
- Figure S1 - S6 (Appendix Figure S1 - S6) show control blots for the IP experiments or Western blots in the main figures. I strongly recommend that these blots are added to the figure panels itself, as an important quality control of expression levels. Only if all information is in one figure panel, a quick comparison of IP efficiency and expression levels or loading is easily possible. As it stands, the source data for these controls is also missing (as it is part of the Appendix).
- Table S1 needs to be renamed to Table EV1.
- Please complete the Author Checklist. Information on manuscript number and author is missing (top left) and Column "Information included in the manuscript?" needs to be filled (drop-down menu).
- Please update the information on antibodies and RNA/DNA sequences (EV tables instead of Appendix).
- Approval of animal experiments by National University of Singapore: please provide the reference or approval number.
- The Methods section should be called Materials and Methods
- Our production/data editors have asked you to clarify several points in the figure legends (see below). Please incorporate these changes in the manuscript and return the revised file with tracked changes with your final manuscript submission:
 - A) The figure legends for figures 3b-d are not provided in a sequential manner (i.e., legend for figure panel 3d is provided before legends 3b-c). This needs to be rectified.
 - B) Please note that in figure 1j; there is a mismatch between the annotated p values in the figure legend and the annotated p values in the figure file that should be corrected.
 - C) Please note that the error bars are not defined in the legends of figures 2c, i; 3c; 6m-p; 7a-d, f, i.
 - D) Please note that scale bar and its definition are missing for figures 7a-e, g-h.
 - E) Please note that the white arrows are not defined in the legends of figures 7a-b.
- I am not sure whether a discrimination between 'main first' and 'co-first' author is possible. We generally assign 'co-first' or to both authors (equal contribution).

- Please shorten the abstract to 175 words and the title to 100 characters (incl. spaces).

- Finally, EMBO Reports papers are accompanied online by A) a short (1-2 sentences) summary of the findings and their significance, B) 2-3 bullet points highlighting key results and C) a synopsis image that is 550x300-600 pixels large (width x height) in PNG or JPG format. You can either show a model or key data in the synopsis image. Please note that the size is rather small and that text needs to be readable at the final size. Please send us this information along with the revised manuscript.

Once you have made these minor revisions, please use the following link to submit your corrected manuscript:

Link Not Available

If all remaining corrections have been attended to, you will then receive an official decision letter from the journal accepting your manuscript for publication in the next available issue of EMBO reports. This letter will also include details of the further steps you need to take for the prompt inclusion of your manuscript in our next available issue.

- On a different note, I would like to alert you that EMBO Press offers a new format for a video-synopsis of work published with us, which essentially is a short, author-generated film explaining the core findings in hand drawings, and, as we believe, can be very useful to increase visibility of the work. This has proven to offer a nice opportunity for exposure i.p. for the first author(s) of the study. Please see the following link for representative examples and their integration into the article web page:

<https://www.embopress.org/doi/full/10.15252/emj.2019103932>

Thank you for your contribution to EMBO reports.

Yours sincerely,

The authors have addressed all minor editorial requests.

Prof. Carlos Ibanez
Peking University
McGovern Institute & School of Life Sciences
Beijing
China

Dear Carlos,

I am very pleased to accept your manuscript for publication in the next available issue of EMBO reports. Thank you for your contribution to our journal.

Kind regards,

Martina
